# Optimized Farmland Mulching Improves Rainfed Maize Productivity by Regulating Soil Temperature and Phenology on the Loess Plateau in China

Shibo Zhang [1,2], Zhenqing Xia [1,2], Guixin Zhang [1,2], Jingxuan Bai [1,2], Mengke Wu [1,2] and Haidong Lu [1,2,*]

1   College of Agronomy, Northwest A&F University, Xianyang 712100, China; z2335570357@163.com (S.Z.); 17854233612@163.com (Z.X.); zgx199869@163.com (G.Z.); bjx990502@163.com (J.B.); 17792031755@163.com (M.W.)
2   Key Laboratory of Biology and Genetic Improvement of Maize in Arid Areas of Northwest Region, Ministry of Agriculture and Rural Affairs, Northwest A&F University, Xianyang 712100, China
*   Correspondence: lhd2042@163.com

**Abstract:** Owing to global warming, continuously increasing the grain yield of rainfed maize is challenging on the Loess Plateau in China. Plastic film mulching has been extensively utilized in dryland agriculture on the Loess Plateau. However, higher topsoil temperatures under film mulch caused rainfed-maize premature senescence and yield loss. Here, we aimed to explore the influence of topsoil temperature driven by novel double mulching patterns on rainfed maize productivity based on the excellent moisture conservation function of plastic film. A maize field experiment was conducted in two different areas, namely Changwu, a typical semi-arid area, and Yangling, a dry semi-humid area. The experiment followed a randomized block design with three replications. Five flat-planting practices were examined in 2021 and 2022: (1) bare land (CK), (2) transparent film mulching (PFM), (3) black film mulching (BFM), (4) double mulching of PFM with a black polyethylene net (PFM + BN), and (5) double mulching of PFM with whole maize stalks (PFM + ST). Soil hydrothermal conditions, maize growth dynamics, grain yield, water use efficiency (WUE), and economic returns were quantified under different mulching practices. Under double mulching treatments, topsoil temperatures were lower than PFM by 1.7–2.0 °C at the two sites ($p < 0.05$), whereas BFM was slightly lower than that of PFM by 0.6–0.7 °C at Yangling ($p > 0.05$). The average growth period for maize under double mulching was longer than that under PFM by 8–11 days at the two sites. Double mulching treatments significantly improved the leaf area index (LAI), chlorophyll relative content (SPAD), and aboveground biomass compared to CK and PFM during the late growth stage. Compared with PFM, average grain yield increased by 14.93%, 18.46%, and 16.45% in Changwu ($p < 0.05$) under BFM, PFM + BN, and PFM + ST, respectively, and by 2.71%, 24.55%, and 20.38% in Yangling. The corresponding WUEs also increased. Additionally, net income under BFM was higher than that under other treatments, and there were no significant ($p > 0.05$) differences between PFM + ST and BFM in Changwu. However, PFM + ST in net income averaged 10.72–52.22% higher than other treatments, and its output value was 19.51% higher in Yangling. In summary, smallholder farmers can adopt PFM + ST to improve rainfed-maize productivity in the Loess Plateau in China.

**Keywords:** double mulching; economic benefits; grain yield; water use efficiency



## 1. Introduction

The growing global population and subsequent rise in food consumption create unprecedented demands for ensuring an adequate grain yield [1]. Addressing this issue is of utmost importance, particularly in the context of climate change and environmental degradation [2]. Dryland agriculture is the primary agricultural production system globally, with the majority of smallholder farmers living in regions where dryland agriculture is practiced [3]. As the world's most populous country, China had a population of 1.40 billion

people in 2019, representing 18% of the global population. The country possesses 128 million hectares of arable land, with approximately 60% relying on rainfed agriculture [4]. The Loess Plateau, spanning an area of $6.4 \times 10^5$ km$^2$ and located between 100–114° E and 33–42° N, extends into the Yellow River basin and serves as the largest rainfed agricultural region. This region provides sustenance for over 100 million individuals [5]. It encompasses 80% of China's total cultivated land area and benefits from ample light and heat resources, making it highly conducive to crop productivity [6]. However, rainfed agriculture is heavily reliant on rainfall, which is susceptible to the impacts of climate change [7,8]. Consequently, these factors often result in reduced crop yields. Therefore, it is imperative to enhance crop productivity within the context of global climate change to ensure food security for the growing population, particularly in dryland agriculture areas.

Plastic film mulching is a crucial cultivation technology that is widely used in dryland agriculture in the Loess Plateau to reduce soil evaporation, enhance soil moisture infiltration, alter soil temperature, and ultimately improve grain yield [9–12]. In China, transparent plastic film coverings are commonly used for agricultural production as a mulching material due to their excellent water retention and low cost [13,14]. Numerous studies have shown that transparent film mulching (PFM) positively affects crop yield compared to bare land. For instance, transparent plastic films increased the yield of wheat, potato, and maize by 10–15% [15], 57–78% [16], and 32–56% [17], respectively. In general, transparent film mulching (PFM) improves crop productivity compared to bare land. Rainfed maize (*Zea mays* L.) is the main cereal crop of the Loess Plateau, cultivated in 27.3% of the total agricultural area [18]. It is almost entirely grown in semi-arid and dry semi-humid areas [17,19–21]. However, numerous studies conducted in warm areas of the Loess Plateau through field experiments have consistently reported the detrimental impact of elevated soil temperature on maize yield when transparent mulch is used [8,22–24]. One well-known mechanism of premature senescence in film-mulched maize is the accelerated growth period caused by the increased temperature of the plastic film. This results in drought stress during the bell-mouthing stage, leading to premature senescence and reduced yield in the later stage. Another recently highlighted mechanism is the intensified absorption of nutrients and water by maize roots in the early growth stage, due to the soil warming effect of plastic film mulch during warmer growing seasons. This intensification leads to reduced availability of moisture and nutrients in the late growth stage, resulting in premature senescence and yield loss [8,22,25]. The third potential mechanism suggests that plastic film mulching improves topsoil temperatures and accelerates maize development during the early growth period. This leads to a decrease in source capacity due to reductions in the leaf area index and photosynthetic rate, as well as a decrease in sink capacity due to fewer kernel numbers. Furthermore, the growth degree of kernel volume is lower, resulting in maize yield loss [24]. While there are controversies regarding the mechanism of premature senescence in maize covered with plastic film, the decrease in maize yield is strongly associated with elevated topsoil temperatures caused by the plastic film. At the same time, as a key indicator influencing maize yield, the increase in soil temperature will shorten the phenological period of crops, ultimately leading to a reduction in maize production [23].

Notably, sole straw mulching, black plastic film mulching, and removing transparent plastic film mulching at later growth stages were used to decrease soil temperature on the Loess Plateau. Sole straw mulching is effective in lowering soil temperatures during warm seasons and mitigating high soil temperatures due to its ability to retain soil moisture from minimal precipitation [26,27]. However, the impact of sole straw mulch on crop yield has been inconsistent, with both increases and decreases reported [28–31]. Another option is the use of black plastic film mulching, which reduces soil temperature by limiting light transmission and radiant heat transfer compared to transparent mulching [32,33]. Several studies have shown that black plastic film mulching leads to higher maize yields compared to transparent plastic film mulching [8,22], although contradictory results have also been observed [34], and no significant difference has been found between the two types of plastic

film [35]. It is worth noting that black film is more expensive than transparent film and may not be readily adopted by smallholder farmers [36]. Additionally, removing the white film during later growth stages has been found to positively impact maize yield [26], but it also results in significant increases in labor costs [24]. According to the Intergovernmental Panel on Climate Change (IPCC), the global average surface temperature increased by 0.74 °C in the 20th century and may further increase by 3–4 °C by the end of this century [37]. Predictions also indicate that to meet the future crop demands of 9.8 billion people without significant changes in the existing cropped land area, a global increase in land use of approximately 100 M ha and a tripling of international trade will be required by 2050 [38]. As a result, the likelihood of maize production shocks will greatly increase due to future warming and population growth [39]. Therefore, it is crucial to regulate soil temperature by improving the mulching system to ensure food security in the face of future warming and population growth. Double mulching with flat cropping cultivation is considered a potential strategy for improving rain-fed maize yield by reducing the topsoil temperature under the film [23]. However, there are still uncertainties regarding the effectiveness of double mulching in improving rainfed maize grain yield at multiple sites. Additionally, the economic returns of double mulching on the Loess Plateau of China have not been reported. The majority of the Loess Plateau is characterized as semi-arid and semi-humid, making it prone to drought. However, it has sufficient light and temperature conditions to enhance crop productivity, making it an optimal area for rainfed maize production in China [6].

Therefore, this study is based on the in-situ testing of two typical ecological regions (semi-arid and dry semi-humid) of dryland agriculture on the Loess Plateau of China in 2021 and 2022. The study aims to investigate the impact of a novel double mulching pattern on rainfed maize productivity while maintaining the soil moisture conservation function of plastic film. Specifically, the objectives of this study are as follows: (1) to analyze the spatial and temporal distribution characteristics of soil hydrothermal conditions under different planting patterns; (2) to examine the effects of double mulching (using black polyethylene net or whole maize stalks) on maize growth dynamics, grain yield, and water use efficiency; and (3) to compare the economic benefits of different mulching patterns.

## 2. Materials and Methods

### 2.1. Site Description

The two-year field experiment was conducted during the spring maize growing seasons of 2021–2022 at two different sites on the Loess Plateau. These sites, Changwu (35°59′ N, 107°38′ E, 1220.0 m elevation) and Yangling (34°20′ N, 108°24′ E, 466.7 m elevation), varied in terms of altitude, temperature, and precipitation (Figure 1A). Changwu is located in a semi-arid area with a mean annual temperature of 9.1 °C, an annual sunshine duration of 2226 h, and a mean annual evaporation of 1500 mm, which is much higher than the annual precipitation of 542 mm. Yangling is in a typical semi-humid and drought-prone area with a mean annual temperature of 12.9 °C, an annual sunshine duration of 2196 h, and a mean annual evaporation of 1440 mm, which is much higher than the annual precipitation of 630 mm. Automated weather observation stations at both experimental sites recorded daily maximum and minimum air temperatures as well as precipitation throughout the two maize growing seasons (Figure 1B). In Changwu, the total precipitation amounts were 302.0 mm (2021) and 288.6 mm (2022), and the mean temperatures were 20.2 °C (2021) and 20.4 °C (2022). In Yangling, the total precipitations were 321.3 mm (2021) and 341.1 mm (2022), and the mean temperatures were 24.4 °C (2021) and 24.5 °C (2022) during the two growing seasons. The groundwater at the experimental sites was below 50 m, and the upward movement of soil water could not reach the root zone; therefore, groundwater can be negligible for maize growth. In accordance with the USDA textural classification system, the soil types of Changwu and Yangling were dark loessial soil and silty clay loam, respectively. The soil properties of the 0–20 cm soil layer measured before the field experiment in 2021 using the recommended methods [40] are shown in Table 1.

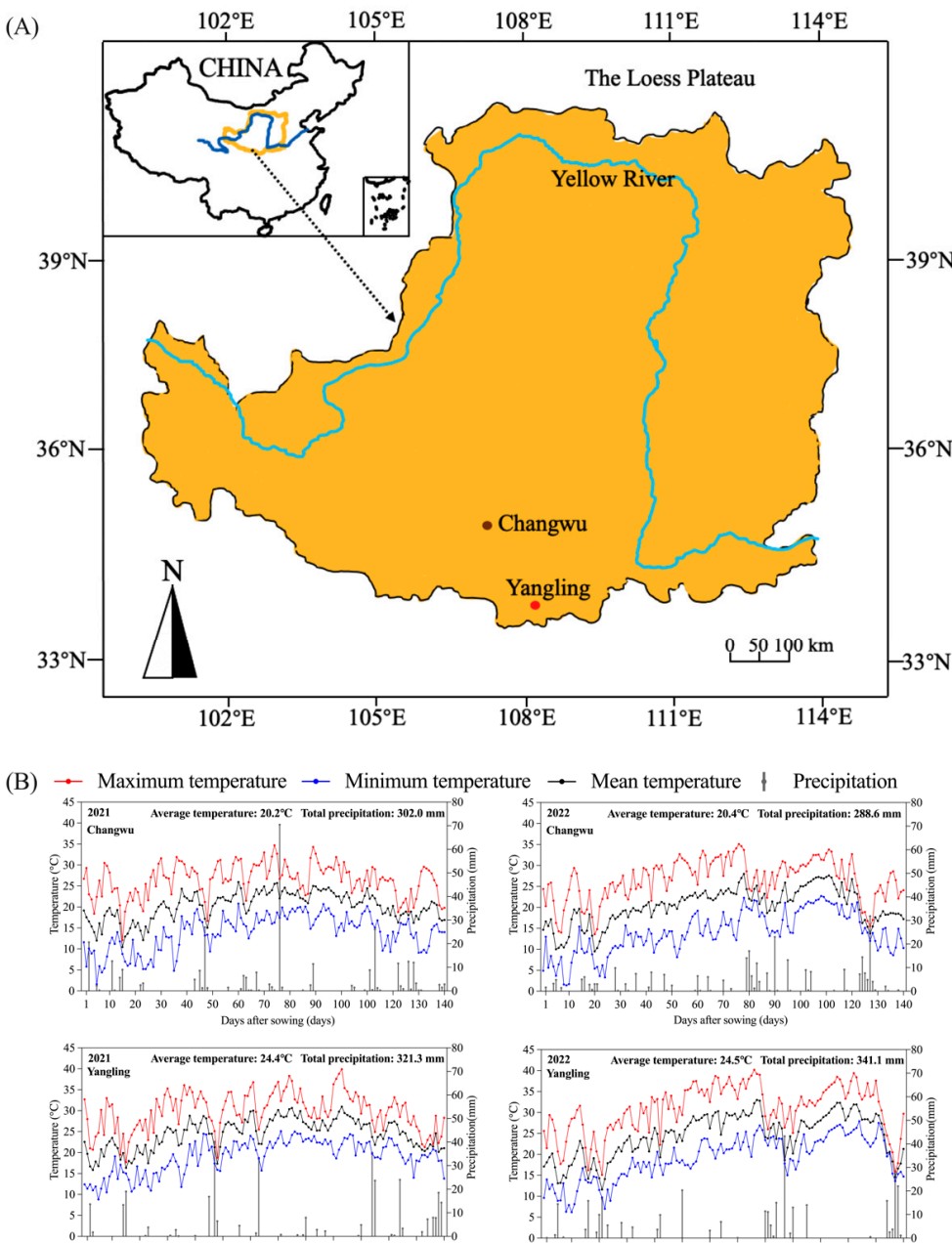

**Figure 1.** The locations of the Changwu and Yangling experiment stations on the Loess Plateau in China (**A**) and the daily meteorological data (including maximum temperature, minimum temperature, mean temperature, and precipitation) during the spring maize growing seasons in 2021 and 2022 at Changwu and Yangling (**B**).

**Table 1.** Physicochemical characteristics of soil in 0–20 cm layer.

| Sites | pH | Soil Organic Matter (g kg$^{-1}$) | Total Nitrogen (g kg$^{-1}$) | Available Nitrogen (mg kg$^{-1}$) | Available Phosphorus (mg kg$^{-1}$) | Available Potassium (mg kg$^{-1}$) |
|---|---|---|---|---|---|---|
| Changwu | 8.5 | 11.34 | 0.98 | 45.11 | 15.32 | 137.98 |
| Yangling | 7.3 | 12.19 | 1.31 | 55.29 | 11.39 | 163.23 |

*2.2. Experimental Design*

A randomized block design experiment was applied in this study. The experiment area was divided into 15 plots with three replicates, including (1) CK, bare land, (2) PFM, transparent plastic film (98% transmittance) mulching, (3) BFM, black plastic film (2%

transmittance) mulching, (4) PFM + BN, double mulching of PFM with black polyethylene net, and (5) PFM + ST, double mulching of PFM with whole maize stalks (whole maize stalk coverage was 3000 kg ha$^{-1}$) (Figure 2). The plastic film used for the film mulching treatments was a polyethylene film with a thickness of 0.008 mm and a width of 0.7 m, and the black nets used for mulch treatment were polyethylene material with a transmittance of 2% and a width of 0.7 m (produced by Zhengzhou Manlv Engineering Materials Co., Ltd., Zhengzhou, China). Small wooden sticks were used as materials to fix the black sunshade net, and plastic ropes were used to tie the wooden sticks and the black net to ensure that the black net and the plastic film did not come into contact and that the black net was located 5 cm above the mulch film. Since the delayed warming of soils in early spring postponed sowing and affected seed germination [41], black nets and maize stalks were mulched at the third leaf stage. All plots were arranged with three replications, and each plot was a flat plot with an area of 33.0 m$^2$ (6.0 m long × 5.5 m wide). A widely planted hybrid maize cultivar, known as "Zhengdan 958", was cultivated with a planting density of 67,500 plants per hectare. The cultivation took place in late April, and the harvest occurred in September. Specifically, at Changwu, the maize was sown on 30 April and harvested on 26 September 2021. In 2022, the maize was sown on 23 April and harvested on 10 September. At Yangling, the maize was sown on 30 April and harvested on 15 September 2021. In 2022, the maize was sown on 30 April and harvested on 4 September. Chemical fertilizers were uniformly applied at a depth of 20 cm in the soil layer, with rates of 225 kg N ha$^{-1}$ and 120 kg P$_2$O$_5$ ha$^{-1}$. This application was done using a plowing-fertilizing integrated machine three days prior to maize planting. Additionally, a mulching machine was used to apply transparent film and black film for mulching one day before maize sowing. Notably, new mulching materials were used each year at both experimental sites; maize straw for mulching was removed at harvest, and it was never integrated into the soil. The maize field was flat, and no irrigation was performed. Additionally, no herbicides were applied during the maize growth periods, and manual weeding was performed when necessary.

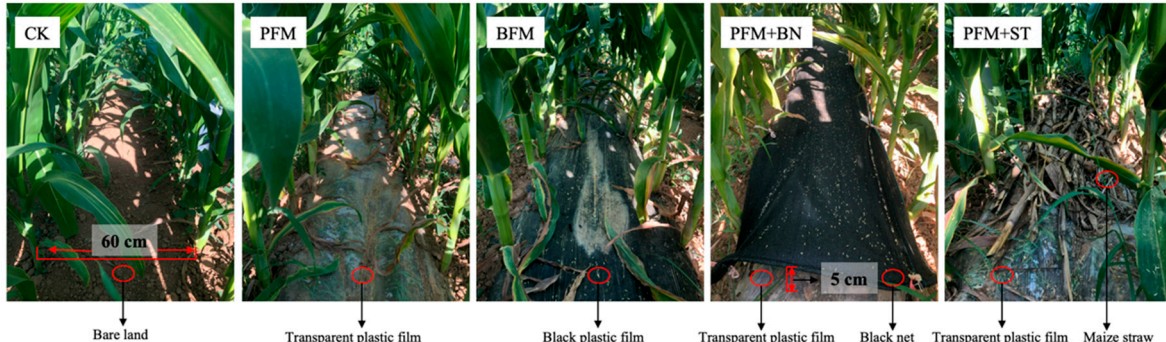

**Figure 2.** Schematic diagram showing the fields conducted with five different treatments. CK indicates bare land; PFM indicates transparent plastic film mulching; BFM indicates black plastic film mulching; PFM + BN indicates double mulching of PFM with black polyethylene net; PFM + ST indicates double mulching of PFM with whole maize stalks. Flat planting with 0.6 m equal row spacing. The black polyethylene net was 5 cm above the transparent plastic film.

### 2.3. Sampling and Measurement

#### 2.3.1. Soil Temperature and Moisture

Every five days after the three-leaf stage (V3), soil temperature at depths of 5, 10, 15, 20, and 25 cm was recorded at 8:00 h, 10:00 h, 12:00 h, 14:00 h, 16:00 h, 18:00 h, and 20:00 h daily with a curved tube mercury geothermometer (Hongxing Thermal Instruments, Hengshui, China) placed between the maize plants in each plot. The mean daily topsoil temperature value at depths from 5 to 25 cm was calculated using daily readings in Changwu and Yangling. Diurnal variation in topsoil temperature at 5 to 25 cm was also determined among different planting patterns.

The percentage of soil water content (SWC) was sampled throughout a sampling depth of 1.0 m in 0.2 m depth increments using a 70 mm diameter portable auger in the middle between two maize rows at sixth leaf (V6), twelfth leaf (V12), silking (R1), grain milking (R3), and physiological maturity (R6) stages for two years in Changwu and Yangling [42]. Samples of SWC were collected at three randomly selected points in each plot. Soil bulk density (BD) was determined for the whole soil profile (0–100 cm depth) and averaged at 1.37 $\text{g cm}^{-3}$ (Changwu, 2021), 1.38 $\text{g cm}^{-3}$ (Changwu, 2022), 1.39 $\text{g cm}^{-3}$ (Yangling, 2021), and 1.41 $\text{g cm}^{-3}$ (Yangling, 2022), respectively. The SWC (%) and soil water storage SWS (mm) were calculated as follows:

$$SWC = [(FW - DW)/DW] \times 100\% \tag{1}$$

$$SWS = SWC \times BD \times SD \times 10/100 \tag{2}$$

where FW and DW were fresh weight (g), and dry weight (g) of each soil sample, respectively. BD is soil bulk density ($\text{g cm}^{-3}$), and SD is soil depth (20 cm).

### 2.3.2. Maize Development Stage Evaluation

The growth stage was recorded when over 50% of maize plants in each plot reached a specific growth stage by a standardized maize development stage system [43], including the vegetative (VS) (i.e., VE: emergence stage, V6: sixth leaf stage, V12: twelfth leaf stage) and reproductive (RS) stages (i.e., R1, silking stage; R3, grain milking stage; R6, physiological maturity stage).

### 2.3.3. Plant Height, Leaf Area Index (LAI), and Chlorophyll Relative Content (SPAD Value)

In each plot, ten adjacent plants were selected for plant height and SPAD value at the sixth leaf stage (V6), twelfth leaf stage (V12), silking stage (R1), grain milking stage (R3), and physiological maturity stage (R6) for two years at the two sites. A portable SPAD 502 (Konica Minolta, Inc., Tokyo, Japan) was used to determine the chlorophyll relative content (SPAD value). Notably, the topmost fully opened leaf was sampled at the V6 and V12 stages, while the maize ear leaf was sampled at the R1, R3, and R6 stages.

At the same time, five maize plants from each plot were randomly sampled at the sixth leaf stage (V6), twelfth leaf stage (V12), silking stage (R1), grain milking stage (R3), and physiological maturity stage (R6) to determine the leaf area (leaf length × leaf width × 0.75) of each maize plant [44]. The leaf area index (LAI) was then calculated using the following equation:

$$LAI = \text{leaf area } (\text{m}^2 \text{ plant}^{-1}) \times \text{plant density (plants ha}^{-1})/10{,}000 \ (\text{m}^2 \text{ ha}^{-1}). \tag{3}$$

### 2.3.4. Aboveground Dry Matter Accumulation

Three adjacent representative maize plants in a row were chosen from each experimental plot to determine aboveground dry matter accumulation at the sixth leaf (V6), twelfth leaf (V12), silking (R1), grain milking (R3), and physiological maturity (R6) stages for two years in Changwu and Yangling, respectively. The harvested plants were dried at 105 °C for 30 min and weighed after oven-drying for 48 h at 80 °C for aboveground dry matter accumulation, and the aboveground dry matter accumulation in each plot was expressed in terms of kg dry matter $\text{ha}^{-1}$ throughout the maize growth stages.

### 2.3.5. Maize Yield Determination

At the physiological maturity stage, maize plants from a 6 m × 3 m site in the middle two rows of each plot were harvested for determination of population level ear number (harvest ears $\text{ha}^{-1}$) and grain yield (kg $\text{ha}^{-1}$) under 14% grain moisture. Among the harvested ears, 30 ears were collected for the number of kernels per ear and a dry weight of 100 kernels. The 100-kernel dry weight (g) was determined by oven-drying 30 samples of 100 kernels to a constant weight at 80 °C for 72 h.

2.3.6. Evapotranspiration (ET), Water Use Efficiency (WUE), and Precipitation Use Efficiency (PUE)

ET (mm) was calculated using the following formula [45]:

$$ET = (SWS_s - SWS_h) + P \tag{4}$$

where $SWS_s$ (mm) is the soil water storage for the 0–100 cm soil depth before sowing, $SWS_h$ (mm) is the soil water storage for the 0–100 cm soil depth at harvesting, P (mm) is the precipitation during the maize growing seasons, and ET (mm) is soil evapotranspiration during the maize growing seasons.

WUE (kg ha$^{-1}$ mm$^{-1}$) and PUE (kg ha$^{-1}$ mm$^{-1}$) were calculated according to the following formulas [35]:

$$WUE = Y/ET \tag{5}$$

$$PUE = Y/P \tag{6}$$

where WUE (kg ha$^{-1}$ mm$^{-1}$) is water use efficiency, PUE (kg ha$^{-1}$ mm$^{-1}$) is precipitation use efficiency, Y is grain yield of spring maize, ET (mm) is soil evapotranspiration during the maize growing seasons, and P (mm) is precipitation during the maize growing seasons. ET (mm) is soil evapotranspiration during the maize growing seasons.

2.3.7. Economic Benefits

The major input sources of consumable items were seeds, fertilizer, labor (including manual seeding and artificial mulching polyethylene black net and straw), machines (including rotation, fertilization, and mulching transparent film and black film), and mulching material (transparent plastic film, black plastic film, and polyethylene black net). Notably, the whole maize straw used as mulching material was harvested from the previous season, so the material cost of the straw was negligible. Input value, output value, and net income were expressed in terms of CNY ha$^{-1}$ and were calculated according to the following formulas [46]:

$$\text{Input value} = \text{labor cost} + \text{machine cost} + \text{seed cost} + \text{fertilizer cost} + \text{mulching materials cost} \tag{7}$$

$$\text{Output value} = \text{maize grain yield} \times \text{maize grain price} \tag{8}$$

$$\text{Net income} = \text{Output value} - \text{Input value} \tag{9}$$

*2.4. Statistical Analysis*

Residual tests were conducted prior to data analysis to ensure that all data met the criteria for normal distribution. SPSS 18.0 (SPSS Inc., Chicago, IL, USA) was used to analysis of variance (ANOVA). The least significant difference (LSD) test ($p < 0.05$) was used to determine differences in the soil water content, mean soil water storage, mean soil temperature, plant height, leaf area index (LAI), SPAD value, aboveground dry matter accumulation, grain yield, kernel number per ear, 1000-kernel dry weight, evapotranspiration (ET), water use efficiency (WUE), precipitation use efficiency (PUE), output value, and net income value. Additionally, a three-way ANOVA was conducted to determine the significant effects of different years, sites, treatments, and their interactions on the plant height, LAI, aboveground dry matter weight, kernel number per ear, 1000-kernel dry weight, grain yield of spring maize, and ET, WUE, and PUE. GraphPad Prism 8 (GraphPad Software Inc., San Diego, CA, USA) was used to draw figures.

## 3. Results

*3.1. Soil Hydrothermal Dynamics*

3.1.1. Soil Moisture

SWC was significantly affected by the different farmland mulching treatments, and it varied spatially and temporally in the semi-arid area (Changwu) and dry semi-humid area

(Yangling) in 2021 and 2022 (Figure 3). SWC in the 0–20 cm soil layer was ranked as follows: PFM > BFM > double mulching treatments (PFM + BN and PFM + ST) > CK, but there was no significant ($p > 0.05$) difference between PFM + BN and PFM + ST. The SWC in the surface layer (0–20 cm) changed sharply, and the SWC in all mulching treatments was significantly ($p < 0.05$) higher than that in the bare land (CK) at the early growth stage of maize. With maize evolution, the SWC of 0–100 cm in all treatments increased or decreased. Specifically, during the early maize growth stages (i.e., at V6 and V12 stages), PFM significantly ($p < 0.05$) decreased SWC in the 20–60 cm soil layers compared to the PFM + BN and PFM + ST, and SWC under BFM was slightly lower than that of the double mulching treatments. In contrast, BFM reduced SWC by 1.82–3.06% and 1.82–3.20% at the 20–60 cm soil layer compared to PFM at R1 and R3 stages in 2021 and 2022, respectively. Similarly, in the 20–60 cm layer at R1 and R3 stages, PFM + BN significantly decreased SWC by an average of 6.57–8.91% and 3.76–9.44% compared to PFM, and by 2.65–7.68% and 5.95–6.96% under PFM + ST, respectively. The SWS dynamics (0–100 cm) were substantially different during the two growing seasons in Changwu and Yangling (Figure 4). In 2021, SWS was highest at the V6 stage and lowest at the R6 stage (Changwu) and R3 stage (Yangling). In 2022, SWS was lowest at the V12 stage (Changwu) and R1 stage (Yangling) and then increased through the later growth stages. At the early growth stages (V6 and V12 stages), there was no significant ($p > 0.05$) difference in SWS among the mulching treatments (except for V6 stage Changwu in 2021 and V12 stage Yangling in 2022). However, when it turned to late growth stages (R1−R6 stage), the SWS under PFM was always significantly ($p < 0.05$) higher than that under PFM + BN and PFM + ST. In general, regardless of treatments, the average SWS was higher in Yangling than in Changwu. Additionally, at both sites, the average SWS under mulching treatment was significantly ($p < 0.05$) higher than that of CK, and there was no significant ($p > 0.05$) difference in SWS among mulching treatments.

### 3.1.2. Soil Temperature

The differences in topsoil temperature varied with different growth stages, and differences in the topsoil temperature among different treatments gradually decreased, especially during the R1–R6 growth stage (Figure 5). At Changwu, the topsoil temperature was ranked during the V3–R6 stage as follows: PFM > BFM> PFM + ST > PFM + BN, and relative to PFM, BFM decreased the topsoil temperature by 1.3 °C, 1.4 °C, 1.1 °C, 1.0 °C, and 1.1 °C on average during maize growth stages (V3–V6, V6–V12, R1–R3, and R3–R6 stages) in two years; double mulching treatments significantly ($p < 0.05$) reduced topsoil temperature by 2.1–2.9 °C, 2.0–2.6 °C, 1.5–2.0 °C, 1.4–1.9 °C, and 1.5–2.0 °C at the corresponding growth stages (Figure 5A). The changing trend of change in soil temperature among treatments at Yangling was similar to that in Changwu, but BFM slightly decreased the soil temperature by 0.7 °C, 0.8 °C, 0.7 °C, 0.5 °C, and 0.5 °C, respectively, compared to PFM among maize growth stages. PFM + BN and PFM + ST treatments decreased soil temperature by 1.6–2.5 °C, 1.5–2.3 °C, 1.7–2.2 °C, 1.8–2.3 °C, and 1.1–1.5 °C (Figure 5B). During the two growing seasons (V3–R6) in Changwu and Yangling, average topsoil temperature was significantly ($p < 0.05$) higher in PFM, BFM, PFM + BN, and PFM + ST than in CK by 3.3 °C, 2.4 °C, 1.1 °C, and 1.6 °C, respectively (Figure 5C). In Changwu, the average topsoil temperature of PFM was significantly ($p < 0.05$) higher than that of other mulching treatments, and there was no significant ($p > 0.05$) difference between the average topsoil temperature of BFM and that of PFM + BN and PFM + ST treatments. However, the average topsoil temperatures of PFM and BFM treatments were significantly ($p < 0.05$) higher than those of double mulching treatments in Yangling (Figure 5C).

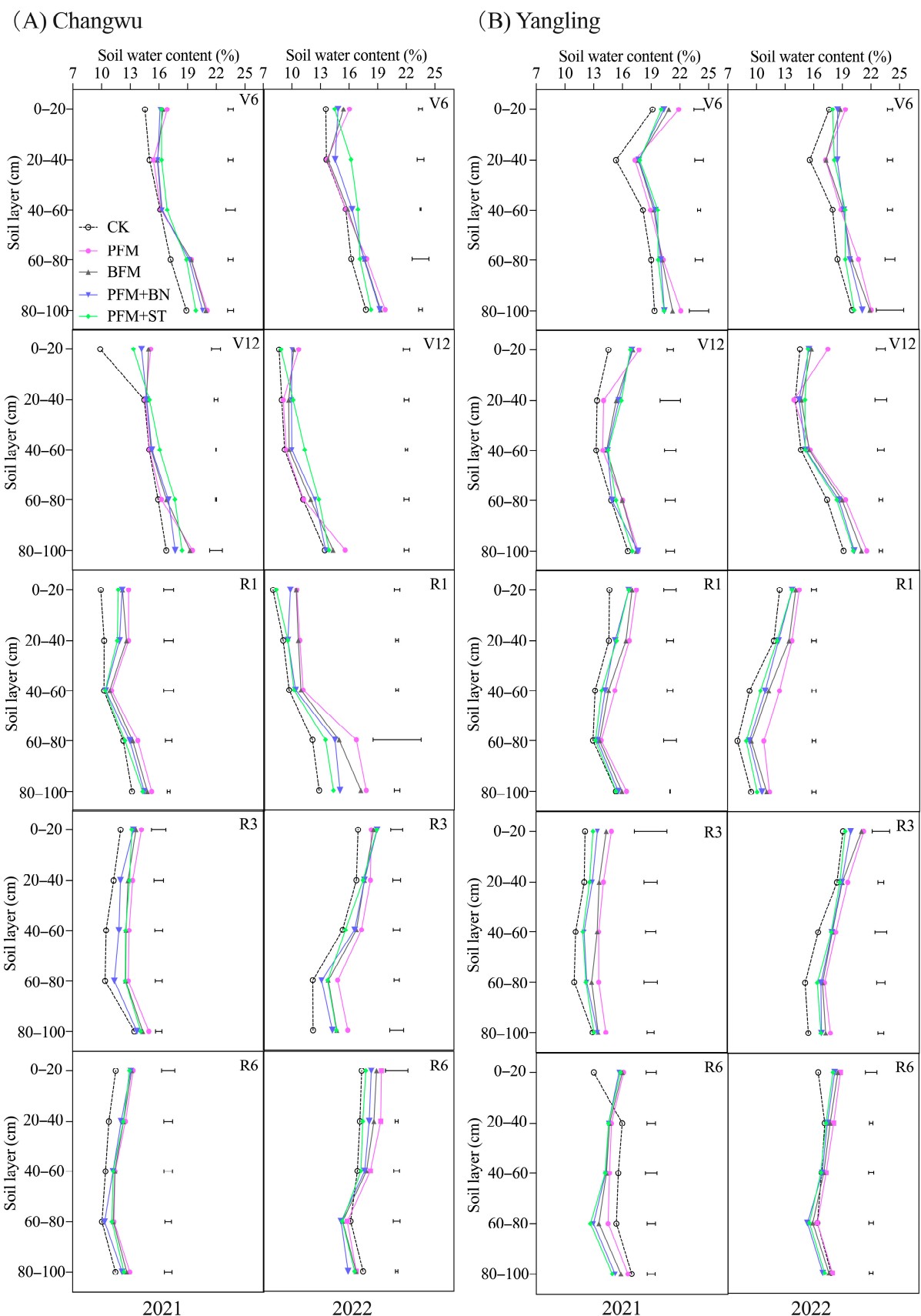

**Figure 3.** SWC in the 0–100 cm under CK (bare land), PFM (transparent plastic film mulching), BFM (black plastic film mulching), PFM + BN (double mulching of PFM with black polyethylene net), and

PFM + ST (double mulching of PFM with straw) during growing seasons in 2021 and 2022 at Changwu (**A**) and Yangling (**B**). Horizontal bars indicate LSD at 0.05 levels. V6, V12, R1, R3, and R6 indicate sixth leaf, twelfth leaf, silking, grain milking, and physiological maturity stages, respectively.

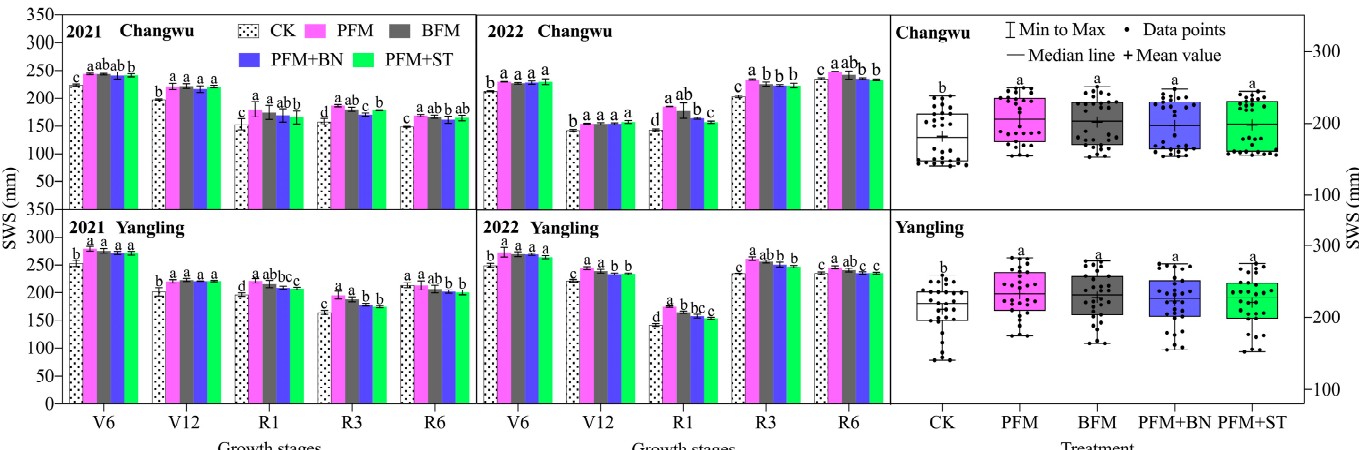

**Figure 4.** SWS in the 0–100 cm during different growing seasons, and average SWS under CK (bare land), PFM (transparent plastic film mulching), BFM (black plastic film mulching), PFM + BN (double mulching of PFM with black polyethylene net), and PFM + ST (double mulching of PFM with straw) in 2021 and 2022 at Changwu and Yangling. Vertical bars stand for one standard error of the mean. The points in the box chart are the data of every growth stage in every cropping year (including 3 replicates, n = 30). Different lowercase letters indicate significant ($p < 0.05$) difference based on LSD test. V6, V12, R1, R3, and R6 indicate sixth leaf, twelfth leaf, silking, grain milking, and physiological maturity stages, respectively.

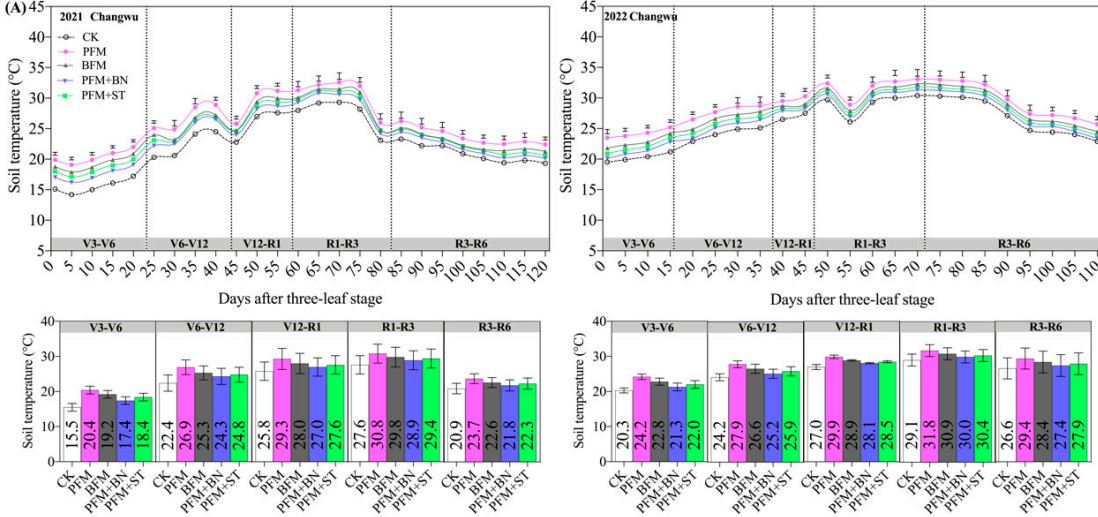

**Figure 5.** *Cont.*

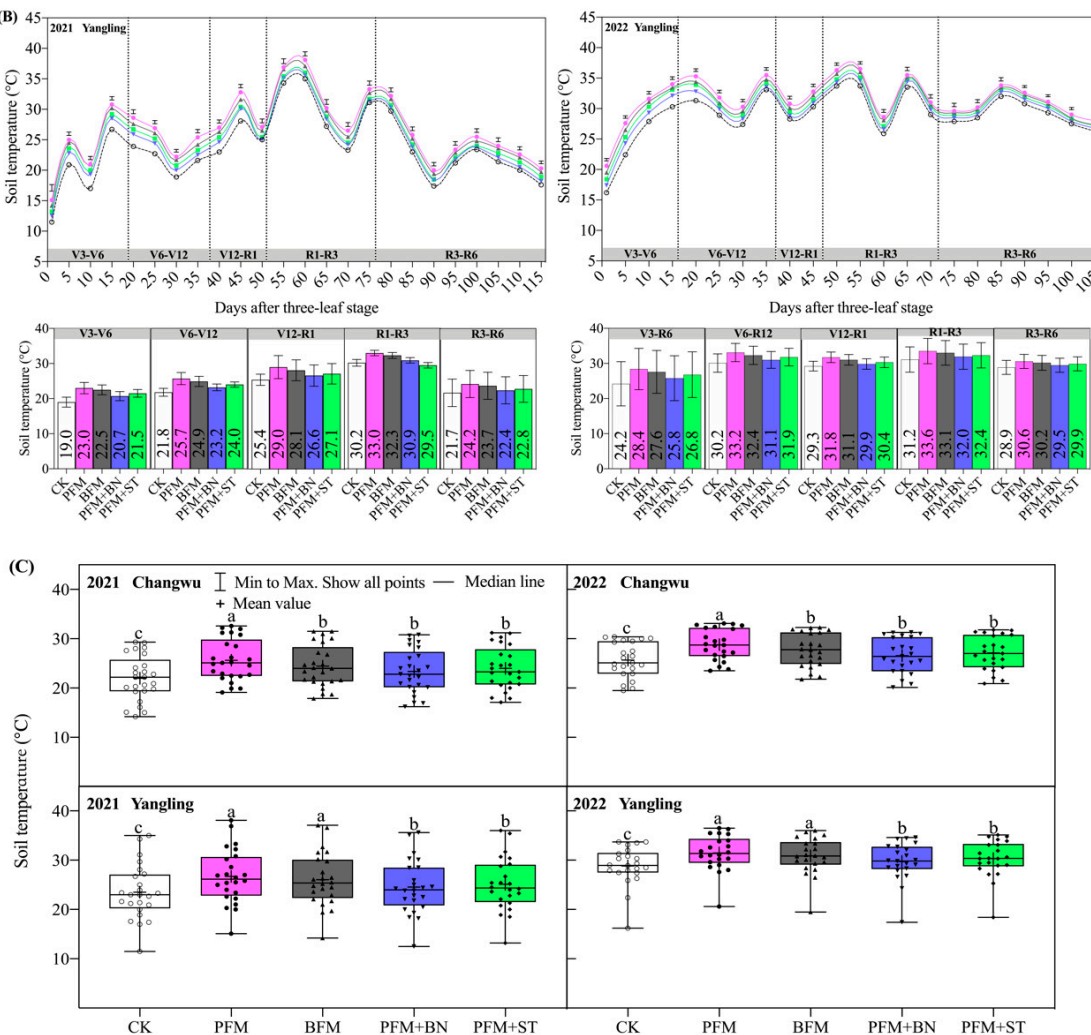

**Figure 5.** Variations in the mean daily temperature during the day (08:00–20:00) at days after three-leaf stage in Chang (**A**) and Yangling (**B**), and average topsoil temperature (**C**) among CK, PFM, BFM, PFM + BN, and PFM + ST at Changwu and Yangling in 2021 and 2022. CK, PFM, BFM, PEM + BN, PFM + ST indicate bare land, transparent plastic film mulching, black plastic film mulching, double mulching of PFM with black polyethylene net, and double mulching of PFM with straw, respectively. V6, V12, R1, R3, and R6 indicate sixth leaf, twelfth leaf, silking, grain milking, and physiological maturity stages, respectively. Vertical bars indicate LSD at 0.05 levels in curve graphs, and vertical bars indicate one standard error of the means. The points in the box chart are the average daily topsoil temperatures for each treatment (n = 25 for Changwu in 2021, n = 23 for t Changwu in 2022, n = 24 for Yangling in 2021, n = 22 for Yangling in 2022). Different lowercase letters indicate significant (*p* < 0.05) difference based on LSD test (**C**).

### 3.2. Maize Phenology

At both sites, development was promoted by the PFM treatment, as the crops advanced to specific vegetative and reproductive stages earlier in the PFM than in CK (e.g., on average, in 2021 and 2022, maize seedling emergence was 2–6 d and 2–3 d earlier at Changwu and Yangling, respectively; maize reached the silking stage 8–10 d and 6–8 d earlier, respectively) (Figure 6). However, there was no significant difference in the duration from sowing to the physiological maturity stage (R6) between the double mulching treatments and CK at Changwu and Yangling in 2021 and 2022. Notably, compared with PFM, BFM on average extended seedling emergence, vegetative growth (VS), reproductive growth (RS), and sowing to physiological maturity by 0–2 d, 2 d, 4–7 d, and 6–9 d at Changwu, but BFM on average prolonged the corresponding maize growth stage by only 0–1 d, 0–1 d,

0–1 d, and 1–2 d at Yangling. In addition, double mulching treatments prolonged maize phenology compared with PFM. Specifically, on average, from 2021 to 2022, VS, RS, and the entire growth period from sowing to physiological maturity under double mulching was prolonged compared to PFM by 2–3 d, 3–5 d, and 9–13 d at Changwu, and 2–3 d, 6–7 d, and 7–9 d at Yangling.

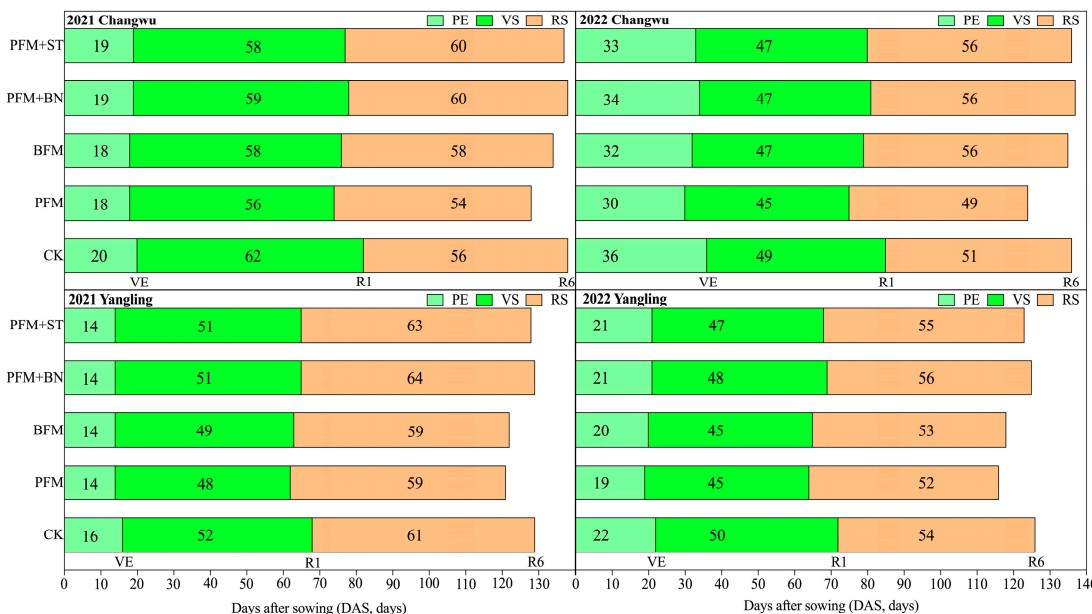

**Figure 6.** Duration of the emergence (PE, from planting to emergence), vegetative stages (VS, from seedling emergence to silking), and reproductive stages (RS, from silking to physiological maturity) of the spring maize under CK (bare land), PFM (transparent plastic film mulching), BFM (black plastic film mulching), PFM + BN (double mulching of PFM with black polyethylene net), and PFM + ST (double mulching of PFM with straw) at Changwu and Yangling in 2021 and 2022.

### *3.3. Maize Growth Dynamics*

#### 3.3.1. Plant Height, Leaf Area Index (LAI), and Chlorophyll Relative Content (SPAD Value)

Site (S) and treatment (T) significantly ($p < 0.05$) affected the height of spring maize plant height (Table 2), and all mulching treatments significantly ($p < 0.05$) increased plant height compared to CK in Changwu and Yangling (Figure 7). The PFM had the tallest plants during early growth (V6 and V12), while the PFM + BN had the tallest plants during later growth (R1 and R3). Importantly, the plant height of Changwu maize was higher than that of Yangling maize at R6 (i.e., 261.6–280.0 cm in 2021 and 260.4–273.9 cm in 2022 at Changwu, and 224.3–251.9 cm in 2021 and 209.0–221.6 cm in 2022 at Yangling).

**Table 2.** Effects of years, sites, and treatments on plant height of spring maize. Y, S, and T indicate year, site, and treatment, respectively. *, $p < 0.05$; **, $p < 0.01$; ***, $p < 0.001$; NS, not significant.

|  | V6 | V12 | R1 | R3 | R6 |
|---|---|---|---|---|---|
| ANOVA |  |  |  |  |  |
| Year (Y) | *** | NS | *** | *** | *** |
| Site (S) | * | *** | *** | *** | *** |
| Treatment (T) | *** | *** | *** | *** | *** |
| Y × S | NS | *** | *** | *** | *** |
| Y × T | * | *** | ** | ** | ** |
| S × T | NS | NS | NS | NS | NS |
| Y × S × T | NS | *** | NS | NS | NS |

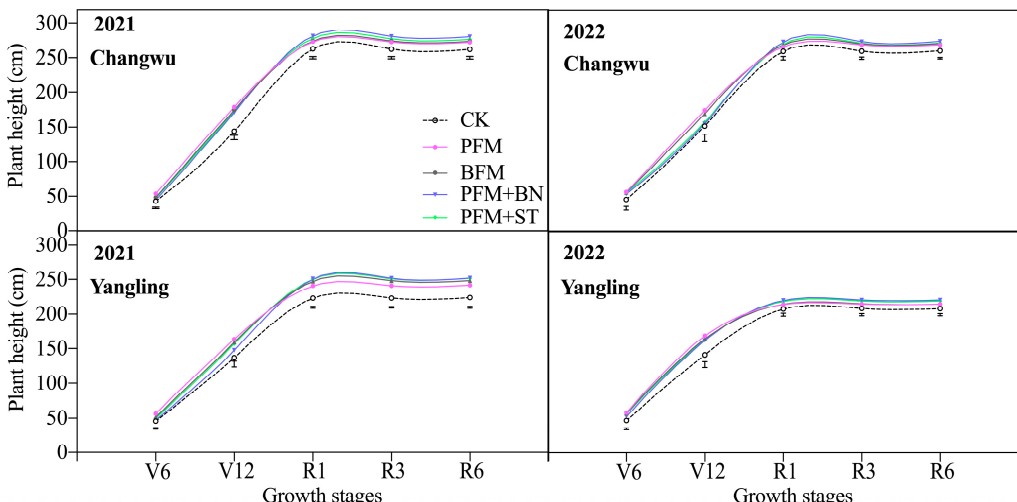

**Figure 7.** Plant height under different treatments at Changwu and Yangling in 2021–2022. CK, PFM, BFM, PEM + BN, PFM + ST indicate bare land, transparent plastic film mulching, black plastic film mulching, double mulching of PFM with black polyethylene net, and double mulching of PFM with straw, respectively. V6, V12, R1, R3, and R6 indicate sixth leaf, twelfth leaf, silking, grain milking, and physiological maturity stages, respectively. Vertical bars indicate LSD at 0.05 levels.

Site (S) and treatment (T) significantly ($p < 0.05$) affected the LAI and SPAD of spring maize (Table 3). The LAI and SPAD values increased dramatically during the vegetative stages, peaked at the R1 stage, and then declined, indicating leaf senescence, in plots subjected to each of the treatments (Figure 8A,B). At the V6 stage, maize plants had very small LAI and SPAD, with no significant ($p > 0.05$) differences in the LAI and SPAD under the mulching treatments. At the V12 stage, under mulching treatments, maize leaves were wide, large, and grew vigorously, PFM had the largest LAI and SPAD, with minimum LAI and SPAD for PFM + BN, and all mulching treatments had no significant ($p > 0.05$) differences for LAI and SPAD. However, significant differences for all mulching treatments were determined at R1, that is, at Changwu, LAIs and SPAD under BFM, PFM + BN, and PFM + ST were higher than those of PFM in 2021 and 2022. At Yangling, the LAIs and SPAD values of double mulching were significantly ($p < 0.05$) higher relative to PFM, and there was no significant ($p > 0.05$) difference between BFM and PFM, especially in 2021. After R1, maize entered the reproductive growth period, and double mulching treatments delayed leaf senescence and maintained the LAI and SPAD at relatively high levels. On the contrary, the PFM tended to cause premature senescence, thus causing the most rapid decline in LAI and SPAD. Meanwhile, there was no significant ($p > 0.05$) difference in LAI and SPAD values between the BFM and PFM at Yangling in 2021. Notably, LAIs and SPAD of all treatments at Changwu were higher than those of the corresponding treatments at Yangling during the maize reproductive growth period.

**Table 3.** Effects of years, sites, and treatments on leaf area index (LAI) and SPAD value of spring maize. Y, S, and T indicate year, site, and treatment, respectively. *, $p < 0.05$; **, $p < 0.01$; ***, $p < 0.001$; NS, not significant.

| | V6 | | V12 | | R1 | | R3 | | R6 | |
|---|---|---|---|---|---|---|---|---|---|---|
| | LAI | SPAD Value | LAI | SPAD Value | LAI | SPAD Value | LAI | SPAD Value | LAI | SPAD Value |
| **ANOVA** | | | | | | | | | | |
| Year (Y) | *** | *** | *** | *** | ** | *** | *** | *** | *** | *** |
| Site (S) | ** | ** | *** | NS | * | NS | *** | * | *** | NS |
| Treatment (T) | *** | *** | *** | ** | *** | *** | *** | *** | *** | *** |
| Y × S | *** | ** | NS | *** | NS | *** | NS | NS | *** | ** |
| Y × T | *** | NS | NS | NS | NS | NS | NS | NS | NS | NS |
| S × T | NS | NS | NS | NS | NS | NS | NS | NS | NS | NS |
| Y × S × T | * | NS | *** | NS | NS | NS | NS | NS | NS | NS |

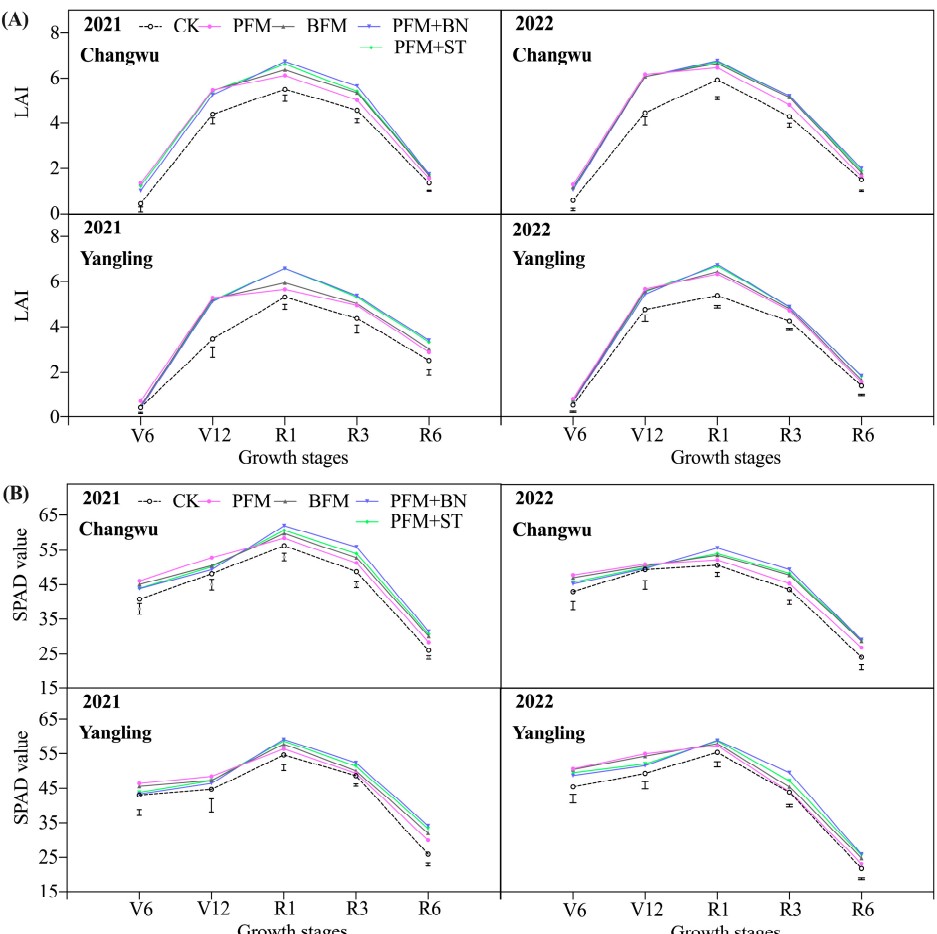

**Figure 8.** Leaf area index (LAI), (**A**), and SPAD value (**B**) under different treatments at Changwu and Yangling in 2021–2022. CK, PFM, BFM, PEM + BN, PFM + ST indicate bare land, transparent plastic film mulching, black plastic film mulching, double mulching of PFM with black polyethylene net, and double mulching of PFM with straw, respectively. V6, V12, R1, R3, and R6 indicate sixth leaf, twelfth leaf, silking, grain milking, and physiological maturity stages, respectively. Vertical bars indicate LSD at 0.05 levels.

### 3.3.2. Aboveground Dry Matter Accumulation

Site (S) and treatment (T) significantly ($p < 0.05$) affected the dry matter accumulation of spring maize (Table 4). Maize dry matter accumulation was consistently significantly ($p < 0.05$) higher under PFM, BFM, PFM + BN, and PFM + ST than under CK during different growth periods in 2021 and 2022 (Figure 9). At the V6 stage, PFM had greater biomass accumulation compared with the BFM, PFM + BN, and PFM + ST. Subsequently, dry matter accumulation increased rapidly in BFM, PFM + BN, and PFM + ST. There was no significant ($p > 0.05$) difference in dry matter accumulation among all mulching treatments at the V12 stage. At the R1 stage, PFM + BN had the largest dry matter accumulation, and PFM + ST had the second largest shoot dry matter accumulation among all mulching treatments, and there was no significant ($p > 0.05$) difference between PFM + BN and PFM + ST. BFM and double mulching treatments had significantly ($p < 0.05$) greater dry matter accumulation than PFM at Changwu, whereas PFM + BN and PFM + ST treatments significantly ($p < 0.05$) increased dry matter accumulation compared to PFM and BFM at Yangling. Similarly, at the R3 stage, the changing trend of dry matter accumulation under all mulching treatments was consistent with that of R1 at Changwu and Yangling. At the R6 stage, PFM + BN increased dry matter accumulation by 1.1–1.5%, 2.2–2.5%, 11.0–12.8%, and 31.5–35.2% relative to PFM + ST, BFM, PFM, and CK at Changwu, respectively, for the two years of the study carried out. Similarly, at Yangling, PFM + BN increased dry matter

accumulation by 1.4–1.7%, 12.2–15.6%, 13.8–20.0%, and 27.5–37.7% compared to PFM + ST, BFM, PFM, and CK, respectively.

**Table 4.** Effects of years, sites, and treatments on dry matter accumulation of spring maize. Y, S, and T indicate year, site, and treatment, respectively. *, $p < 0.05$; **, $p < 0.01$; ***, $p < 0.001$; NS, not significant.

|  | V6 | V12 | R1 | R3 | R6 |
|---|---|---|---|---|---|
| ANOVA |  |  |  |  |  |
| Year (Y) | ** | *** | *** | *** | *** |
| Site (S) | *** | *** | *** | *** | *** |
| Treatment (T) | *** | *** | *** | *** | *** |
| Y × S | NS | *** | NS | NS | NS |
| Y × T | * | NS | NS | NS | NS |
| S × T | *** | NS | NS | NS | ** |
| Y × S × T | *** | NS | NS | NS | NS |

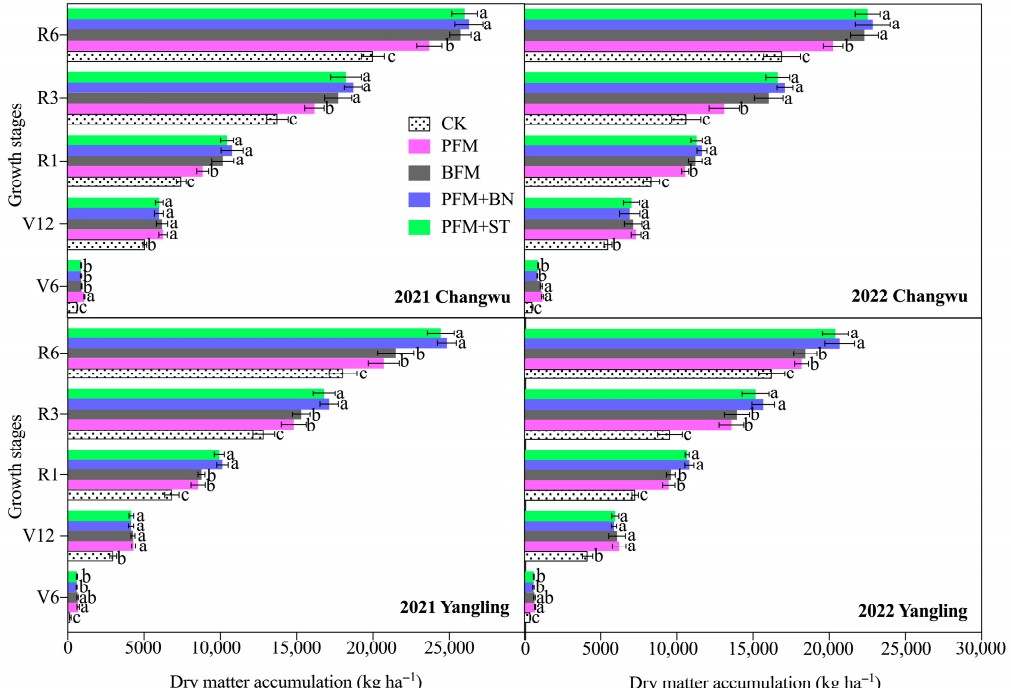

**Figure 9.** Dry matter accumulation under different treatments at Changwu and Yangling in 2021–2022. CK, PFM, BFM, PEM + BN, PFM + ST indicate bare land, transparent plastic film mulching, black plastic film mulching, double mulching of PFM with black polyethylene net, and double mulching of PFM with straw, respectively. V6, V12, R1, R3, and R6 indicate sixth leaf, twelfth leaf, silking, grain milking, and physiological maturity stages, respectively. Different lowercase letters indicate significant differences when $p < 0.05$.

### 3.4. Yield Components and Grain Yield

Site (S) and treatment (T) significantly ($p < 0.05$) affected the kernel number per year, 1000-kernel dry weight, and grain yield of spring maize (Table 5). At both experimental sites, kernel number per ear under PFM, BFM, PFM + BN, and PFM + ST treatments was significantly ($p < 0.05$) higher than that of CK (Figure 10A), but no significant ($p > 0.05$) difference was observed for harvest ear number (Figure S1). In particular, there were significant ($p < 0.05$) differences in 1000-kernel dry weight under mulching treatments (Figure 10B). On average, the 1000-kernel dry matter under BFM, PFM + BN, and PFM + ST was significantly ($p < 0.05$) higher than that of PFM by 6.29–9.36%, 9.74–11.84%, and 8.56–10.40%, respectively, at Changwu. Conversely, at Yangling, there was no significant ($p > 0.05$) difference in the 1000-kernel dry weight of BFM and PFM, and the 1000-kernel dry

weights of BFM and PFM were significantly ($p < 0.05$) lower than those of PFM + BN and PFM + ST. Compared with PFM, PFM + BN, and PFM + ST significantly ($p < 0.05$) increased 1000-kernel dry weight by 14.09% and 11.95% (Yangling, 2021) and 14.64% and 11.54% (Yangling, 2022), respectively. PFM + BN and PFM + ST significantly ($p < 0.05$) increased by 10.87% and 8.80% (Yangling, 2021) and 13.04% and 9.99% (Yangling, 2022) relative to BFM. In general, the ear number harvested, kernel number per ear, and 1000-kernal dry weight of Changwu were higher than those of Yangling in 2021 and 2022, especially the kernel number per ear and 1000-kernal dry weight. Significant differences ($p < 0.05$) were observed between double mulching and PFM for grain yield at the two sites in 2021 and 2022 (Figure 10C). At Changwu, the grain yields of PFM + BN and PFM + ST were significantly ($p < 0.05$) higher than that of PFM by 13.97–15.81% and 18.93–21.11% in 2021 and 2022, respectively, and by 22.24–24.45% and 18.52–24.65% in Yangling.

**Table 5.** Effects of years, sites, and treatments on kernel number per ear, 1000-kernel dry weight, and grain yield of spring maize. Y, S, and T indicate year, site, and treatment, respectively. **, $p < 0.01$; ***, $p < 0.001$; NS, not significant.

|  | Kernel Number Per Ear | 1000-Kernel Dry Weight | Grain Yield |
|---|---|---|---|
| ANOVA |  |  |  |
| Year (Y) | *** | *** | *** |
| Site (S) | *** | *** | *** |
| Treatment (T) | *** | *** | *** |
| Y × S | NS | *** | *** |
| Y × T | NS | *** | NS |
| S × T | *** | *** | *** |
| Y × S × T | ** | ** | NS |

### 3.5. ET, WUE, and PUE

Year (Y), site (S), and treatment (T) significantly ($p < 0.05$) affected the ET, WUE, and PUE (Table 6). ET under all treatments showed no significant difference ($p > 0.05$), but different mulching patterns significantly ($p < 0.05$) affected WUE and PUE, and the performance differed between the two sites. On average, from 2021 to 2022, at Changwu, compared with CK, mulching treatments significantly increased the WUEs by 25.29%, 42.02%, 51.47%, and 47.00%, respectively, and PUE increased by 25.64%, 44.27%, 48.63%, and 46.11%. Similarly, on average from 2021 to 2022, the WUEs under PFM, BFM, PFM + BN, and PFM + ST were higher than that of CK by 23.21%, 26.22%, 56.03%, and 50.59%, respectively, and were 19.29%, 22.63%, 48.57%, and 43.77% higher for PUE at Yangling. Notably, compared with PFM, BFM increased WUE and PUE in Yangling by 2.45% and 2.80%, respectively, but increased the average WUE and PUE in Changwu by 13.36% and 14.83%. Furthermore, double mulching treatments increased WUE and PUE at Changwu by 47.00–51.47% and 46.11–48.63% relative to PFM, and increased the average WUE and PUE at Yangling by 50.59–56.03% and 43.77–48.57%. WUE and PUE under BFM were significantly ($p < 0.05$) higher than those of PFM at Changwu. PFM + BN and PFM + ST obtained higher WUE and PUE at Changwu compared with those at Yangling.

### 3.6. Economic Benefits

The data revealed that spring maize produced input costs and economic benefits achieved by different planting patterns (Figure 11). The input cost in the CK plots was markedly lower than that in mulching systems under the same fertilizer and seed input costs, and the input cost of CK was 23.74%, 26.13%, 49.59%, and 31.83% lower than that of PFM, BFM, PFM + BN, and PFM + ST, respectively. Additionally, the input cost difference between the mulching systems was mainly reflected in labor (including seeding and film setting) and mulching material costs (Figure 11A). Nevertheless, the PFM + BN has the highest output value, especially in Yangling, which was significantly ($p < 0.05$) higher than other mulching treatments; then the PFM + ST took second place, while there was no

significant ($p > 0.05$) difference under the output value between the PFM + ST and BFM at Changwu in 2021 (Figure 11B). Notably, the net income of all mulches was significantly ($p < 0.05$) higher than that of bare land by 19.51–44.54% at Changwu and Yangling in 2021 and 2022 (Figure 11C). At Changwu, although the net income of the BFM treatment was higher than that of PFM + ST, there was no significant ($p > 0.05$) difference between BFM and PFM + ST. The average net income of PFM + ST in two years was 9.18%, 20.89%, and 23.87%, which was significantly higher than that of PFM + BN, BFM, and PFM at Yangling.

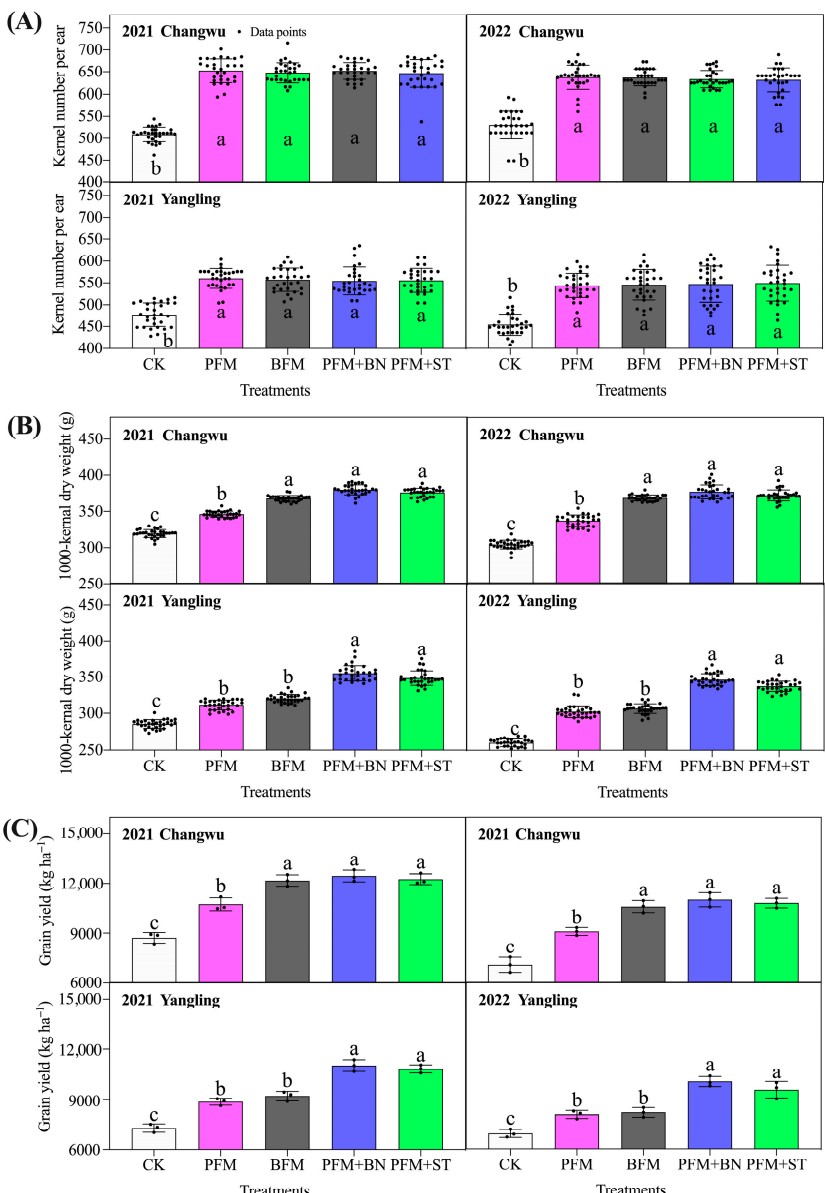

**Figure 10.** Kernel number per ear (**A**), 1000-kernel dry weight (**B**), and grain yield (**C**) under different treatments at Changwu and Yangling in 2021–2022. CK, PFM, BFM, PEM + BN, PFM + ST indicate bare land, transparent plastic film mulching, black plastic film mulching, double mulching of PFM with black polyethylene net, and double mulching of PFM with straw, respectively. Different lowercase letters indicate significant differences when $p < 0.05$. Black dots represent replicates for each treatment (n = 30 for kernel number per ear, n = 30 for 1000-kernel dry weight, n = 3 for grain yield).

**Table 6.** Effects of sites and treatments on Total evapotranspiration (ET), water use efficiency (WUE), and precipitation use efficiency (PUE) in 2021–2022. CK, PFM, BFM, PEM + BN, and PFM + ST indicate bare land, transparent plastic film mulching, black plastic film mulching, double mulching of PFM with black polyethylene net, and double mulching of PFM with straw, respectively. Y, S, and T indicate year, site, and treatment, respectively. *, $p < 0.05$; **, $p < 0.01$; ***, $p < 0.001$; NS, not significant (in A). Different lowercase letters indicate significant differences when $p = 0.05$.

| Year | Site | Treatment | ET (mm) | WUE (kg ha$^{-1}$ mm$^{-1}$) | PUE (kg ha$^{-1}$ mm$^{-1}$) |
|---|---|---|---|---|---|
| 2021 | Changwu | CK | 324.94 ± 10.38 [ab] | 27.09 ± 1.39 [c] | 28.87 ± 1.06 [c] |
| | | PFM | 312.14 ± 3.09 [b] | 33.16 ± 1.43 [b] | 35.60 ± 1.29 [b] |
| | | BFM | 322.04 ± 4.88 [ab] | 36.49 ± 2.16 [ab] | 40.31 ± 1.15 [a] |
| | | PFM + BN | 324.28 ± 5.85 [ab] | 39.89 ± 1.12 [a] | 41.23 ± 1.16 [a] |
| | | PFM + ST | 334.06 ± 10.17 [a] | 37.72 ± 0.99 [a] | 40.57 ± 1.07 [a] |
| | Yangling | CK | 344.82 ± 9.59 [a] | 21.14 ± 0.64 [c] | 22.68 ± 0.77 [c] |
| | | PFM | 334.04 ± 6.42 [ab] | 26.58 ± 1.10 [b] | 27.61 ± 0.64 [b] |
| | | BFM | 329.61 ± 4.71 [b] | 27.98 ± 1.2 [b] | 28.68 ± 0.89 [b] |
| | | PFM + BN | 327.72 ± 8.07 [b] | 33.92 ± 0.95 [a] | 34.37 ± 1.07 [a] |
| | | PFM + ST | 325.58 ± 6.21 [b] | 33.11 ± 1.02 [a] | 33.75 ± 0.73 [a] |
| 2022 | Changwu | CK | 312.45 ± 9.34 [a] | 22.51 ± 1.37 [c] | 24.55 ± 1.59 [c] |
| | | PFM | 313.92 ± 6.79 [a] | 28.98 ± 0.66 [b] | 31.52 ± 0.82 [b] |
| | | BFM | 314.74 ± 8.61 [a] | 33.95 ± 1.03 [a] | 36.76 ± 1.26 [a] |
| | | PFM + BN | 312.65 ± 4.64 [a] | 35.24 ± 1.23 [a] | 38.17 ± 1.47 [a] |
| | | PFM + ST | 307.44 ± 5.64 [a] | 35.19 ± 0.86 [a] | 37.49 ± 0.99 [a] |
| | Yangling | CK | 362.63 ± 10.21 [a] | 19.11 ± 0.19 [c] | 20.31 ± 0.71 [c] |
| | | PFM | 348.30 ± 12.00 [a] | 23.00 ± 1.48 [b] | 23.66 ± 0.72 [b] |
| | | BFM | 348.97 ± 9.50 [a] | 22.82 ± 1.12 [b] | 24.03 ± 0.86 [b] |
| | | PFM + BN | 351.63 ± 12.74 [a] | 28.87 ± 1.54 [a] | 29.50 ± 0.90 [a] |
| | | PFM + ST | 359.63 ± 10.79 [a] | 27.50 ± 1.49 [a] | 28.05 ± 1.47 [a] |
| | ANOVA | | | | |
| | Year (Y) | | ** | *** | *** |
| | Site (S) | | *** | *** | *** |
| | Treatment (T) | | * | *** | *** |
| | Y × S | | *** | NS | NS |
| | Y × T | | NS | NS | NS |
| | S × T | | NS | *** | *** |
| | Y × S × T | | ** | * | NS |

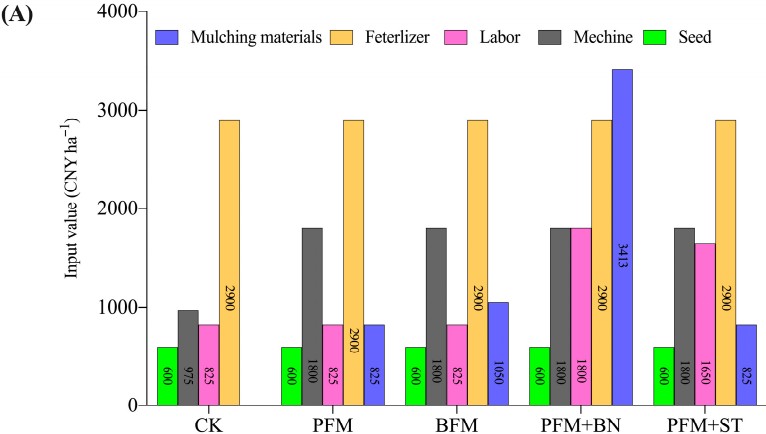

**Figure 11.** *Cont.*

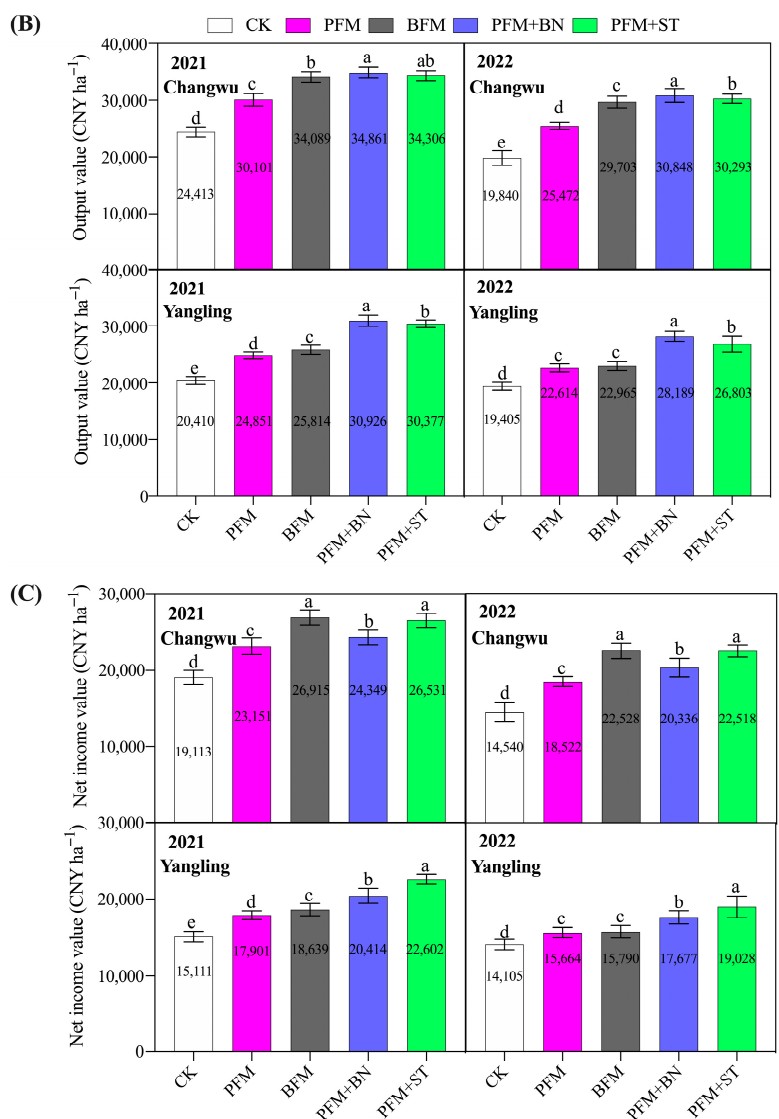

**Figure 11.** Economic input (**A**), output (**B**) and net income (**C**) of the CK (bare land), PFM (transparent plastic film mulching), BFM (black plastic film mulching), PFM + BN (double mulching of PFM with black polyethylene net), and PFM + ST (double mulching of PFM with straw) for spring maize in 2021 and 2022 at Changwu and Yangling. Vertical bars stand for one standard error of the mean. Different lowercase letters indicate significant (*p* < 0.05) differences based on the LSD test.

## 4. Discussion

### 4.1. Soil Hydrothermal Conditions

In rainfed agricultural systems, rainfall is commonly the only source of soil moisture [47]. Therefore, it is crucial to conserve an adequate amount of precipitation in the soil to enhance crop yield. Farmland mulching can enhance the collection of rainfall and increase soil water content (SWC) in the semi-arid and dry semi-humid regions of the Loess Plateau [10,11]. In our study, SWC under farmland mulching treatments (PFM, BFM, PFM + BN, and PFM + ST) was significantly (*p* < 0.05) higher than no-mulching (CK) during maize growing seasons (Figure 3). This is likely because the no-mulching treatment leads to greater soil moisture evaporation losses, whereas mulching can significantly decrease soil moisture evaporation caused by the direct influence of solar radiation. Double mulching treatments during the early stages of maize growth (V6 and V12 stages) significantly increased SWC in the 20–60 cm soil profile compared to the PFM. This effect was particularly pronounced under the PFM + ST in Changwu. In contrast, the PFM had higher SWC than other mulching treatments at the 60–100 cm soil layer during the middle and later stages

of maize growth (R1, R3, and R6 stages). The observed phenomena are likely a result of PFM significantly enhancing maize aboveground transpiration during the early growth stage, leading to increased soil moisture consumption at a depth of 40–60 cm. During the middle and later stages of maize growth, the double mulching treatments (PFM + BN and PFM + ST) necessitated increased moisture for robust maize growth, resulting in the consumption of soil moisture at a depth of 60–100 cm. Notably, mulching practices in Changwu and Yangling significantly ($p < 0.05$) increased average SWS at the 0–100 cm depth compared with CK during growing seasons, but the effects were somewhat different for the mulching treatments at each growth stage (Figure 4). Interestingly, at the maize maturity stage in Yangling, a typically dry semi-humid area, the SWS of the 0–100 cm soil layer under CK was higher than that of the dual-mulching treatments (2021), or there was no significant ($p > 0.05$) difference between CK and dual-mulching treatments (2022) (Figure 4). These occurrences are likely because the rainfall amount is large at maize physiological maturity and soil moisture becomes saturated as infiltration progresses, thereby leading to double mulching treatments that were not conducive to soil moisture infiltration. As demonstrated by previous observations, covering the soil surface reduces the soil moisture infiltration of pepper and onion under soil moisture saturation [48]. Overall, the mulching system in semi-arid areas is more conducive to the soil moisture conservation of 0–100 cm soil moisture than in arid and semi-humid areas, and double mulching systems have a positive effect on improving soil moisture compared to CK in the spring maize growth seasons. Nevertheless, with globally increasing atmospheric aridity over the 21st century [49], soil moisture increase should be based on farmland mulching practices in rainfed areas on the Loess Plateau.

Temperature has a critical effect on crop production, and soil temperature is the basis for changing the temperature received by plants and is of great significance to farmland productivity [50–52]. At Changwu and Yangling, the soil temperatures of sole film mulching treatments (PFM and BFM) were significantly ($p < 0.05$) higher than those of bare land (CK), while the soil temperature under BFM was lower than that under PFM, especially in Changwu (Figure 5A,B). These results are consistent with previous studies on the Loess Plateau [8,22]. This result can be attributed to the different covering materials; the sole plastic mulch prevents water exchange between the air and soil, so the heat exchange and potential heat flux between soil and air are reduced [53], and it can capture solar energy and prevent solar heat loss as a result of reradiation from the soil surface to the surrounding air [54]. On the contrary, black plastic films can absorb most of the solar radiation [55] and decrease soil temperature more than transparent plastic films [22,56]. Previously, some pioneers found that soil temperature with transparent plastic mulch was higher than that with sole straw mulching and that soil temperature with straw mulch was lower than that in a non-mulched field [57]. In this study, we further found that PFM + ST significantly ($p < 0.05$) decreased the average topsoil temperature compared to PFM by 1.7 °C and 1.4 °C, but significantly ($p < 0.05$) increased soil temperature relative to uncovered plots (CK) by 1.8 °C and 1.6 °C at Changwu and Yangling, respectively (Figure 5C). This may be because straw residues mulched on the soil surface typically form a barrier against soil evaporation, ensuring the maintenance of soil moisture and reducing soil temperature [58,59]. In parallel, PFM + BN was observed to significantly ($p < 0.05$) reduce the average topsoil temperature more than PFM by 2.3 °C and 2.0 °C, whereas it significantly ($p < 0.05$) increased the average topsoil temperature more than CK by 1.2 °C and 1.0 °C at Changwu and Yangling, respectively (Figure 5C). The reason for this is likely that black plastic nets can reduce the absorption of solar radiation, thereby decreasing the ambient temperature [60,61]. These views have been confirmed in our study: straw and black polyethylene nets are used as shielding materials to mitigate solar radiation, thereby reducing topsoil temperatures. Notably, there was no significant ($p > 0.05$) difference between PFM + BN and PFM + ST in the two sites, while the average topsoil temperature under double mulching practices (PFM + BN and PFM + ST) were all lower than BFM in Changwu ($p > 0.05$) and Yangling ($p < 0.05$), because the average air temperature of Yangling was 4.2 °C higher than that of

Changwu during the spring maize growing seasons. Additionally, when the maize plant canopy was established, the warming in the topsoil temperature produced by mulching was reduced in magnitude as the vigorous canopy intercepted additional solar radiation. This result is in line with that observed by previous researchers [8,23,25]. Overall, the double mulching treatments moderately reduced the soil temperature compared to the single black and transparent films.

*4.2. Maize Phenology, and Growth Dynamics*

Crop phenology is one of the most reliable biological indicators to identify climate and environmental changes, and is usually used to judge climatic trends [62,63]. In particular, crop phenology is affected by both climatic factors and anthropogenic management practices [64]; therefore, when climate factors cannot be changed, the regulation of phenology through anthropogenic management practices is the dominant position. However, the impact of human management practices on crop phenology is spatially variable and usually depends on the planting systems [65]. In this study, the maize growth period under PFM was shortened by 8–12 d relative to CK (Figure 6). This result is in line with previous research [8,22]. Meanwhile, we found that the duration of reproductive growth (RS) and the whole growth period duration under BFM was longer than that under PFM by 4–5 d and 6–9 d in Changwu (Figure 6), as observed in previous studies [8,22]. However, there were no marked differences in the duration of vegetative growth (VS) and RS between BFM and PFM in Yangling. The main reason for this phenomenon may be that Yangling was warmer than Changwu during the growth period of maize, which led to a large increase in soil temperature under BFM, thereby leading to a similar duration of maize growth under BFM and PFM. Notably, there was no significant difference between the CK and double mulching treatments for the duration of the entire growth period at these two experimental sites. Surprisingly, compared with PFM, double mulching treatments prolonged the VS and RS periods of maize by 2–3 d and 5–6 d, respectively, and then extended the whole growth period by 9–10 d in Changwu and Yangling (Figure 6). These results may be due to the soil cooling effect driven by the double mulching system. Therefore, in warm regions, the double mulching treatments prolonged the growth of maize, which may be more conducive to delaying the senescence of maize and the increase in dry matter at the late growth stage.

Plant height, LAI, and SPAD value are induced by component whole-crop-level physiological processes that occur during various developmental phases in the life cycle of the plant, which determine maize grain yield [66]. In particular, aboveground dry matter accumulation is the basis on which maize yield is calculated, and increasing dry matter accumulation is the main strategy to improve maize yield [67]. In our study, PFM and BFM had significantly ($p < 0.05$) greater plant height, LAI, SPAD, and aboveground dry matter accumulation than CK at the early growth stage (Figures 7–9). Similar results have been reported previously [8,35]. During the late growth stage (R1, R3, and R6), the agronomic parameters mentioned above under BFM were significantly ($p < 0.05$) higher than those under PFM treatment in Changwu, while there were no significant ($p > 0.05$) differences in Yangling. In this study, we further found that the agronomic parameters under novel mulching practices (PFM + BN and PFM + ST) were significantly ($p < 0.05$) higher than those under the PFM treatment during the late growth stage in Changwu and Yangling. These results may be attributed to the idea that PFM caused a higher soil temperature, leading to a sharp increase in soil water and nutrient consumption in the early growth period of spring maize compared to double mulching practices, which led to insufficient soil nutrient and water supply at the late growth stage and thereby significantly reduced agronomic parameters [8,22–24]. Although variations in agronomic parameters (plant height, LAI, SPAD value, and aboveground biomass) were similar under the PFM + BN and PFM + ST treatments, the agronomic parameters of the PFM + ST treatment were slightly higher than those of the PFM + BN treatment at the early growth stage, and the agronomic parameters of the PFM + ST treatment were slightly lower than those of the PFM + BN treatment at the late growth stage. Thus, prolonging the growth period of spring maize and

delaying leaf senescence may also increase the interception of light by the canopy, further increasing the aboveground dry matter accumulation after the R1 stage, which is consistent with previous studies [68]. Overall, double mulching (PFM + BN and PFM + ST) improved plant height, LAI, SPAD value, and aboveground biomass during the late growth stage of spring maize.

### 4.3. Maize Grain Yield, WUE, and PUE

The harvest ears, kernel number per ear, and 1000-kernel dry weight are the main components of maize yield. Plastic film mulching can improve yield composition (i.e., harvest ears, kernel number per ear, and 1000-kernel dry weight) and increase grain yield in numerous crops [15–17]. A majority of research on maize film mulching has used transparent film in China [69–71]. Our study found that the grain yield produced by transparent plastic film mulching was significantly ($p < 0.05$) lower than that produced by black plastic film mulching in Changwu (Figure 10), as observed by flat planting with sole transparent film mulching (relative to flat planting with sole black film mulching), and transparent plastic film mulching on ridges (relative to black plastic film mulching on the ridges) significantly ($p < 0.05$) decreased maize grain yield in a typically warm temperate semi-arid climate zone [8,22]. However, at Yangling, there was no significant ($p > 0.05$) difference in maize grain yield between the two types of sole plastic film mulching. This result was consistent with previously reported findings for potatoes in a typical continental temperate climate zone [72]. These findings may occur because the average temperatures in Yangling during the maize growing season were higher than those in Changwu, thereby weakening the cooling effect of the black film (relative to the transparent plastic film). Surprisingly, this study found that double mulching practices significantly increased maize grain yields compared to transparent plastic film mulching in Changwu and Yangling. As summarized previously, a moderate reduction in soil temperature can improve maize yield, especially in warm areas [51]. WUE and PUE are important parameters for measuring sustainable agricultural development [23,73]. Farmland mulching can significantly ($p < 0.05$) improve WUE and PUE in some semi-arid and dry semi-humid regions of the Loess Plateau [16,23]. We also demonstrated that mulching practices increased WUE and PUE compared to CK in Changwu and Yangling in 2021–2022 (Table 6). The main reason is that mulching practices are effective in reducing water loss from soil evaporation, enhancing canopy transpiration, and promoting biomass accumulation, leading to increased grain yield [8,21,22]. ET consists of soil evaporation and plant transpiration, with plant transpiration being recognized as the primary component [74]. Previous studies have confirmed that the difference in ET between mulching and no-mulching was closely associated with precipitation during the maize growing season in semi-arid and dry semi-humid regions [23,75]. Our findings indicate that the evapotranspiration (ET) rates of the mulching practices at both sites were lower than those of the CK, potentially due to higher soil evaporation compared to plant transpiration (Table 6). Grain yield is a crucial factor that determines the WUE and PUE of rainfed maize. In this study, PFM + BN and PFM + ST resulted in higher grain yields and ultimately caused higher WUE and PUE under double mulching treatments in semi-arid and dry semi-humid regions in 2021–2022 (Table 6). In conclusion, the mulching practices have the potential to serve as an effective rainfed agricultural management system on flat plots, enhancing maize yield, WUE, and PUE in the mulched farmland of the Loess Plateau.

### 4.4. Economic Benefits

The economic benefit of maize is a critical scale index that responds to the input and output of the studied system based on marketing value. It is closely associated with cultivator enthusiasm and thus cannot be ignored in the process of optimizing high-yield production patterns [76]. Input values of labor, machines, and mulching materials under mulching systems were higher than CK, while net income was significantly ($p < 0.05$) higher than CK (Figure 11). Many researchers have found that although the input value of farmland mulching systems is higher than that of bare land, the net income of farmers has

greatly increased due to the higher maize yield generated by farmland coverage [8,77,78]. As we all know, smallholder farmers are the main maize growers in China (i.e., there are 22.86 million small farms, and each household manages 0.40 hectares of agricultural land on average) [79], so higher net income makes it easier for farmers to adopt new coverage systems. In this study, we further found that the net income of the black plastic film mulching treatment (BFM) was higher than that of the novel mulching system (PFM + ST), and there was no significant ($p > 0.05$) difference between BFM and PFM + ST at Changwu. At Yangling, the net income of PFM + ST was significantly ($p < 0.05$) higher than that of the other mulching systems (Figure 11). Overall, in rainfed agricultural areas on the Loess Plateau, PFM + ST, as a brand-new covering system, increases economic benefits.

## 5. Conclusions

In semi-arid and dry semi-humid areas of the Loess Plateau with adequate accumulated temperature in China, farmland mulching systems have significantly ($p < 0.05$) improved the soil hydrothermal conditions and ultimately increased rainfed maize grain yield and water use efficiency compared to bare land. In particular, the innovative double mulching system had a higher topsoil temperature compared to bare land, whereas the double mulching system moderately reduced topsoil temperature compared to sole transparent film mulching. The double mulching system effectively delayed premature senescence and significantly increased maize yield and water use efficiency compared to transparent film mulching in semi-arid and dry semi-humid areas. However, there was no statistically significant difference ($p > 0.05$) between the double mulching system and solely black film mulching in dry, semi-humid areas. Nevertheless, global warming exacerbates global economic inequality, and small-holder farmers have a strong preference for the economic benefits of farmland mulching systems. Net income under sole black film mulching was higher than that under other treatments, and there was no significant ($p > 0.05$) difference between double mulching of transparent film with whole maize stalks and black film mulching in semi-arid areas. Yet, double mulching of transparent film with whole maize stalks in net income was 10.72–52.22% averaged higher than other treatments, and 19.51% averaged higher when output value in dry semi-humid areas. Therefore, this study provides a necessary mulching strategy for improving rainfed maize productivity as a high-yield and high-efficiency cultivation technology.

**Supplementary Materials:** The following supporting information can be downloaded at: https://www.mdpi.com/article/10.3390/agronomy13112790/s1. Figure S1, Effects of treatments on harvest ear of spring maize at Changwu and Yangling in 2021–2022. CK, PFM, BFM, PEM + BN, PFM + ST indicate bare land, transparent plastic film mulching, black plastic film mulching, double mulching of PFM with black polyethylene net, and double mulching of PFM with straw, respectively. Black dots represent replicates for each treatment (n = 3). Different lowercase letters indicate significant differences when $p < 0.05$.

**Author Contributions:** Conceptualization, H.L. and S.Z.; methodology, S.Z.; software, S.Z.; validation, S.Z.; formal analysis, S.Z.; investigation, S.Z., Z.X., G.Z., J.B., M.W. and H.L.; resources, S.Z.; data curation, G.Z. and S.Z.; writing—original draft preparation, S.Z.; writing—review and editing, H.L. and S.Z.; visualization, S.Z.; supervision, H.L.; project administration, H.L.; funding acquisition, H.L. All authors have read and agreed to the published version of the manuscript.

**Funding:** This work was supported by funds from the National Natural Science Foundation of China (31771724) and the Key Research and Development Project of Shaanxi Province (2022NY-197).

**Data Availability Statement:** The datasets supporting the results presented in this manuscript are included within this article.

**Conflicts of Interest:** The authors declare no conflict of interest.

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
