# Peer review of "Optimized Farmland Mulching Improves Rainfed Maize Productivity by Regulating Soil Temperature and Phenology on the Loess Plateau in China"

_agronomy, doi:10.3390/agronomy13112790_

Round 1
Reviewer 1 Report
Comments and Suggestions for Authors
The manuscript entitled "Optimized farmland mulching improves rainfed maize productivity by regulating soil temperature and phenology on the Loess Plateau in China" is dealing with different kind of mulch application techniques on soil surface properties and maize crop growth responses. Maize crop growth responses in terms of agronomic parameters and crop water use efficiency have been elaborated in this study. Two sites and two year growth measurements at different crop growth stages have been taken into consideration. Double mulching including transparent white sheet followed by maize stalk have been found as a suitable approach for conserving soil water content, improving water use and precipitation use efficiency, improving economic benefits and crop yield. The study is interesting and findings are sound. The study falls under the scope of the journal. However, there are a few major concern which need to be addressed before proceeding further. The introduction section can be updated. Methods can be elaborated. Statistical analysis section and results presented in table and figures are not in line. Particularly, in the analysis it has been mentioned that the data were pooled during the analysis (ANOVA) but in the results data are presented separately for site-wise, year-wise and growth stage-wise parameters. Also results can be better presented. Some multivariate analyses can be included for better presenting the data. Results section can be reduced by 30% by highlighting important findings only. Some of the figures can be presented as Supporting Information. A few explanations seems ambiguous which can be rephrased for better clarity. See the attached file for detailed comments. The manuscript needs major revisions before further consideration based on the comments annotated in the pdf file.

Comments on the Quality of English LanguageA few sentences need to be modified and a few typos can be curated during the time of revision.
Author Response
Reviewer #1:
The manuscript entitled "Optimized farmland mulching improves rainfed maize productivity by regulating soil temperature and phenology on the Loess Plateau in China" is dealing with different kind of mulch application techniques on soil surface properties and maize crop growth responses. Maize crop growth responses in terms of agronomic parameters and crop water use efficiency have been elaborated in this study. Two sites and two year growth measurements at different crop growth stages have been taken into consideration. Double mulching including transparent white sheet followed by maize stalk have been found as a suitable approach for conserving soil water content, improving water use and precipitation use efficiency, improving economic benefits and crop yield. The study is interesting and findings are sound. The study falls under the scope of the journal. However, there are a few major concern which need to be addressed before proceeding further. The introduction section can be updated. Methods can be elaborated. Statistical analysis section and results presented in table and figures are not in line. Particularly, in the analysis it has been mentioned that the data were pooled during the analysis (ANOVA) but in the results data are presented separately for site-wise, year-wise and growth stage-wise parameters. Also results can be better presented. Some multivariate analyses can be included for better presenting the data. Results section can be reduced by 30% by highlighting important findings only. Some of the figures can be presented as Supporting Information. A few explanations seems ambiguous which can be rephrased for better clarity. See the attached file for detailed comments. The manuscript needs major revisions before further consideration based on the comments annotated in the pdf file.
Responses to Reviewer #1: Thank you very much for your recognition of this study. We have updated the introduction, elaborated on the data statistical methods, changed the charts using the three-factor statistical method, and importantly, reduced the results section. We have carefully revised the manuscript based on your comments. As follows:
Comment 1: Line 37: replace words already present in title with other relevant keywords! also arrange the keywords in alphabetical order!
Response 1: Thank you very much for your helpful comments. We have modified Keywords according to your request. Please see Line 37. As follows: Keywords: Double mulching; economic benefits; grain yield; water use efficiency.
Comment 2: Line 122: present some literature review on phenology!
Response 2: Thank you very much for your helpful comments. We have presented literature on phenology in Discussion section. Please see Line 610-616. As follows:Crop phenology is one of the most reliable biological indicators to identify climate and environmental changes, and is usually used to judge climatic trends [62, 63]. In particular, crop phenology is affected by both climatic factors and anthropogenic management practices [64]; therefore, when climate factors cannot be changed, the regulation of phenology through anthropogenic management practices is the dominant position. However, the impact of human management practices on crop phenology is spatially variable and usually depends on the planting systems [65].
[62] Richardson, A.; Keenan, T.; Migliavacca, M.; Ryu, Y.; Sonnentag, O.; Toomey, M. Climate Change, Phenology, and Phenological Control of Vegetation Feedbacks to the Climate System. Agric. For. Meteorol. 2013, 169, 156–173, doi:10.1016/j.agrformet.2012.09.012.
[63] White, M.A.; de Beurs, K.; Didan, K.; Inouye, D.; Richardson, A.; Jensen, O.; Magnuson, J.; O’Keefe, J.; Zhang, G.; Nemani, R.; et al. Intercomparison, Interpretation, and Assessment of Spring Phenology in North America Estimated from Remote Sensing for 1982-2006. Glob. Chang. Biol. 2008, 5, 10-15.
[64] Yazdanpanah, M.; Azadi, Y.; Mahmoudi, H. Understanding Smallholder Farmers’ Adaptation Behaviors through Climate Change Beliefs, Risk Perception, Trust, and Psychological Distance: Evidence from Wheat Growers in Iran. J. Environ. Manage. 2019, doi:10.1016/j.jenvman.2019.109456.
[65] Hu, X.; Huang, Y.; Sun, W.; Yu, L. Shifts in Cultivar and Planting Date Have Regulated Rice Growth Duration under Climate Warming in China since the Early 1980s. Agric. For. Meteorol. 2017, 247, 34–41, doi:10.1016/j.agrformet.2017.07.014.
Comment 3: Line 157: Table 1 PH
Response 3: Thank you very much for your helpful comments. We have modified “PH” to “pH”. Please, see Table 1.
Comment 4: Line 238: ha−1
Response 4: Thank you very much for your helpful comments. We have modified “ha−1” to “ha−1”. Please, see Line 240.
Comment 5: In 2.4 Statistical analysis: but in figures, data for both the sites are presented separately?
check the statistical analysis and represent it clearly!
Response 5: Thank you very much for your helpful comments. As you are worried, we have rewritten the statistical method and redrawn the figures. Please, see Line 271-282, and please see Figure 8, 9 ,10, 11, 12 and Table 2. As follows:“Residual tests were conducted prior to data analysis to ensure that all data met the criteria for normal distribution. SPSS 18.0 (SPSS Inc., Chicago, IL, USA) was used to analysis of variance (ANOVA). The least significant difference (LSD) test (p < 0.05) was used to determine differences in the soil water content, mean soil water storage, mean soil temperature, plant height, leaf area index (LAI), SPAD value, aboveground dry matter accumulation, grain yield, kernel number per ear, 1000-kernel dry weight, evapotranspiration (ET), water use efficiency (WUE), precipitation use efficiency (PUE), output value and net income value. Additionally, three-way ANOVA was conducted to determine significant effects of different years, sites, treatments, and their interactions on the plant height, LAI, aboveground dry matter weight, kernel number per ear, 1000-kernel dry weight and grain yield of spring maize, and ET, WUE and PUE. GraphPad Prism 8 (GraphPad Software Inc., San Diego, CA, USA) was performed to draw figures.”
Comment 6: Line 300: R1stage
Response 6: Thank you very much for your helpful comments. We have modified “R1stage” to “R1 stage”. Please, see Line 304.
Comment 7: Line 323: need not to include all the details observed, only focus on key findings and highlight/discuss the novelty of the work in the discussion section. follow this for all the results part as the manuscript is already very bulky!
Response 7: Thank you very much for your helpful comments. We have made every effort to reduce the presentation of the results analysis section in accordance with your request. Importantly, we also cut down on the references.
Comment 8: Figure 6: what is the relevance of daily temperature variations? is it linked to phenological variations? authors are advised to present these elaborate information in Supporting information!
Response 8: Thank you very much for your helpful comments. Considering that the existing data can already reflect the differences in topsoil temperature, we have removed the daily soil temperature changes and analyzed the results.
Comment 9: Figure 11: Harvest year number? if statistically not significant, present in a table or as Supplementary Information!
Response 9: Thank you very much for your helpful comments. According to your request, we have presented the number of harvested ears in the attachment.
Comment 10: Figure 12: It would be appreciated to include some multivariate analysis results such as Multiple regressions or Principle Component Analysis for best representing the data and reducing the data volume to reach an appropriate conclusion of the study!
Response 10: Thank you very much for your helpful comments. Figure 12 is a comparison of the economic benefits, which can clearly illustrate the economic differences between different coverage treatments. It is important to note that economic benefits are not the focus of the study in this manuscript. Our research focuses on optimizing cover models to mitigate agronomic traits and yield of maize premature aging.
Comment 11: Line 610-611: “These views have been confirmed in our study that straw and black polyethylene nets are used as shielding materials to mitigate solar radiation, thereby reducing topsoil temperatures.”
black polythene sheets may absorb more light and lead to increase in surface temperature, I think? Author need to rethink on this explanation?
Response 11: Thank you very much for your helpful comments. In our research, we have elaborated in detail in the material method section that the black polyethylene mesh is placed 5cm above the transparent film, and it is important not to come into direct contact with the transparent film. Therefore, the black polyethylene mesh has the effect of blocking sunlight and reducing temperature in the dual mulching treatment.
Comment 12: Line 610-611: the explanations given here can be re-visited!
Response 12: Thank you very much for your helpful comments. At your suggestion, we have removed this explanation. Please see Line 577-578.
Comment 13: Line 751: “Consequently, the double mulching system delayed premature senescence and increased maize yield and water use efficiency more than transparent film mulching in semi-arid and dry semi-humid areas, but there was no significant (p > 0.05) difference between the double mulching system and sole black film mulching in dry semi-humid areas.”
the sentence need to be rephrased for better clarity. it seems ambiguous as semi-humid areas are repeated two times with different observations in a same sentence!
Response 13: Thank you very much for your helpful comments. According to your comments, we have revised this paragraph to make it clearer. Please see 727-731. As follows:
The double mulching system effectively delayed premature senescence and significantly increased maize yield and water use efficiency compared to transparent film mulching in semi-arid and dry semi-humid areas. However, there was no statistically significant difference (p > 0.05) between the double mulching system and sole black film mulching in dry semi-humid areas.
Reviewer 2 Report
Comments and Suggestions for Authors
An interesting study based on 2 years of research in two locations. The aim of the research was to evaluate the spatial and temporal distribution characteristics of soil hydrothermal conditions under different planting patterns, depending on the type of soil mulching in maize cultivation. The advantage of the study is the large number of parameters assessed.
Detailed notes:
1. China's population data should be updated to 2019 instead of 2023
2. Whether traditional or biodegradable plastic film was used. Traditional plastic film causes problems when removing it and remains in the environment
3. Weather data should be related to multi-year data
4. How can the low phosphorus content in the soil be explained?
5. What was the corn previous crop?
6. What is the practical importance of a facility combining plastic film mulch and the use of mulch from corn plants. Is collecting maize straw from the field, storing it and then reusing it not too labor-intensive and limited to small areas. It is difficult to agree with the authors of the work that this is also a costless factor (element) in the economic assessment.
7. The authors estimated the mass of 100 grains - it is common to give the mass of 1000 grains.
8. In references, the titles of cited works are written in capital letters, e.g. item 3 - "Improving Crop Productivity and Resource.... "; It is common to write "Improving crop productivity and resource ..." in lower case.
Author Response
Reviewer #2:
An interesting study based on 2 years of research in two locations. The aim of the research was to evaluate the spatial and temporal distribution characteristics of soil hydrothermal conditions under different planting patterns, depending on the type of soil mulching in maize cultivation. The advantage of the study is the large number of parameters assessed.
Responses to Reviewer #2: Thank you very much for your recognition of this study. We have carefully revised the manuscript based on your comments.
Comment 1: China's population data should be updated to 2019 instead of 2023
Response 1: Thank you very much for your helpful comments. As you are worried, we have made modifications to this issue. Please see Line 46. As follows: As the world’s most populous country, China had a population of 1.40 billion people in 2019, representing 18% of the global population.
Comment 2: Whether traditional or biodegradable plastic film was used. Traditional plastic film causes problems when removing it and remains in the environment.
Response 2: Thank you very much for your helpful comments. As you are worried, the use of traditional and degradable mulching can pose certain environmental pollution risks in the arid areas of the Loess Plateau where small farmers reside, the use of mulching is necessary enhance crop yield and economic benefits improving soil moisture storage. our article focuses on optimizing coverage mitigate the effects of premature aging in maize. The impact of plastic film on environmental pollution an intriguing and labor-intensive topic. We will address this important issue in future research.
Comment 3: Weather data should be related to multi-year data
Response 3: Thank you very much for your helpful comments. We fully agree with your comment. Our meteorological data comes from the local meteorological bureau and experimental stations' meteorological data from 2000 to 2020. The meteorological data we present is very close to the ones in the following references, so you can refer to it. As follows:Changwu is located in a semi-arid area with a mean annual temperature of 9.1 °C, annual sunshine duration of 2226 h, and mean annual evaporation of 1500 mm, which is much higher than the annual precipitation of 542 mm (Bu et al., 2014; Chen et al., 2015; Dai et al., 2021; Liu et al., 2016; Wang et al., 2011). Yangling is in a typical semi-humid and drought-prone area with a mean annual temperature of 12.9 °C, annual sunshine duration of 2196 h, and mean annual evaporation of 1440 mm, which is much higher than the annual precipitation of 630 mm (Dong et al., 2018; Hu et al., 2020; Liu et al., 2017; Saddique et al., 2020; Wang et al., 2023).
Bu, L., Liu, J., Zhu, L., Luo, S., Chen, X., Li, S., 2014. Attainable yield achieved for plastic film-mulched maize in response to nitrogen deficit. Eur. J. Agron. 55, 53–62. https://doi.org/10.1016/j.eja.2014.01.002
Chen, Y., Liu, T., Tian, X., Wang, X., Li, M., Wang, S., Wang, Z., 2015. Effects of plastic film combined with straw mulch on grain yield and water use efficiency of winter wheat in Loess Plateau. F. Crop. Res. 172, 53–58. https://doi.org/10.1016/j.fcr.2014.11.016
Dai, Z., Hu, J., Fan, J., Fu, W., Wang, H., Hao, M., 2021. No-tillage with mulching improves maize yield in dryland farming through regulating soil temperature, water and nitrate-N. Agric. Ecosyst. Environ. 309, 107288. https://doi.org/10.1016/j.agee.2020.107288
Dong, Q., Yang, Y., Yu, K., Feng, H., 2018. Effects of straw mulching and plastic film mulching on improving soil organic carbon and nitrogen fractions, crop yield and water use efficiency in the Loess Plateau, China. Agric. Water Manag. 201, 133–143. https://doi.org/10.1016/j.agwat.2018.01.021
Hu, Y., Ma, P., Duan, C., Wu, S., Feng, H., Zou, Y., 2020. Black plastic film combined with straw mulching delays senescence and increases summer maize yield in northwest China. Agric. Water Manag. 231, 106031. https://doi.org/10.1016/j.agwat.2020.106031
Liu, Q., Chen, Y., Liu, Y., Wen, X., Liao, Y., 2016. Coupling effects of plastic film mulching and urea types on water use efficiency and grain yield of maize in the Loess Plateau, China. Soil Tillage Res. 157, 1–10. https://doi.org/10.1016/j.still.2015.11.003
Liu, Z., Chen, Z., Ma, P., Meng, Y., Zhou, J., 2017. Effects of tillage, mulching and N management on yield, water productivity, N uptake and residual soil nitrate in a long-term wheat-summer maize cropping system. F. Crop. Res. 213, 154–164. https://doi.org/10.1016/j.fcr.2017.08.006
Saddique, Q., Liu, D.L., Wang, B., Feng, P., He, J., Ajaz, A., Ji, J., Xu, J., Zhang, C., Cai, H., 2020. Modelling future climate change impacts on winter wheat yield and water use : A case study in Guanzhong Plain , northwestern China. Eur. J. Agron. 119, 126113. https://doi.org/10.1016/j.eja.2020.126113
Wang, M., Liu, Z., Zhai, B., Zhu, Y., Xu, X., 2023. Long-term straw mulch underpins crop yield and improves soil quality more efficiently than plastic mulch in different maize and wheat systems. F. Crop. Res. 300, 109003. https://doi.org/10.1016/j.fcr.2023.109003
Wang, Y., Shao, M., Zhu, Y., Liu, Z., 2011. Impacts of land use and plant characteristics on dried soil layers in different climatic regions on the Loess Plateau of China. Agric. For. Meteorol. 151, 437–448. https://doi.org/10.1016/j.agrformet.2010.11.016
Zhang, S., Zhang, G., Xia, Z., Wu, M., Bai, J., 2022. Optimizing plastic mulching improves the growth and increases grain yield and water use efficiency of spring maize in dryland of the Loess Plateau in China. Agric. Water Manag. 271, 107769. https://doi.org/10.1016/j.agwat.2022.107769
Comment 4: How can the low phosphorus content in the soil be explained?
Response 4: Thank you very much for your helpful comments. As you are worried, the data for available phosphorus in Table 1 is correct. We have listed some literature to confirm the correctness of our data. We have compiled some literature on local experimental sites to support the effective phosphorus content in Table 1.
Zhang, S., Zhang, G., Xia, Z., Wu, M., Bai, J., 2022. Optimizing plastic mulching improves the growth and increases grain yield and water use efficiency of spring maize in dryland of the Loess Plateau in China. Agric. Water Manag. 271, 107769. https://doi.org/10.1016/j.agwat.2022.107769
Huang, F., Zihan, L., Zhang, P., Jia, Z., 2021. Hydrothermal effects on maize productivity with different planting patterns in a rainfed farmland area. Soil Tillage Res. 205, 104794. https://doi.org/10.1016/j.still.2020.104794
Comment 5: What was the corn previous crop?
Response 5: Thank you very much for your helpful comments. In the two experimental sites of this study, the previous crop of spring maize is also spring maize.
Comment 6: What is the practical importance of a facility combining plastic film mulch and the use of mulch from corn plants. Is collecting maize straw from the field, storing it and then reusing it not too labor-intensive and limited to small areas. It is difficult to agree with the authors of the work that this is also a costless factor (element) in the economic assessment.
Response 6: Thank you very much for your helpful comments. In most cases, maize straw resources (approximately 70% is discarded or burned) are in excess in dryland areas of northern China (Liu et al., 2008). Secondly, we cover corn stalks with transparent plastic film to moderately reduce topsoil temperature, thus alleviating early senescence of spring maize and ultimately achieving higher grain yield. In adopting this coverage strategy on relatively large land, it is inevitable to incur certain artificial costs. The most important aspect of our research is how to solve the agronomic issue of early senescence of spring maize caused in a single transparent film covering system. The focus is not on economic benefits.
Liu, H., Jiang, G., Zhuang, H.Y., Wang, K., 2008. Distribution, utilization structure and potential of biomass resources in rural China: With special references of crop residues. Renew. Sustain. Energy Rev. 12, 1402–1418. https://doi.org/10.1016/j.rser.2007.01.011
Comment 7: The authors estimated the mass of 100 grains - it is common to give the mass of 1000 grains.
Response 7: Thank you very much for your helpful comments. Based on your comment, we have changed the weight of 100 grains to 1000 grains and have presented it in the figure 10C.
Comment 8: In references, the titles of cited works are written in capital letters, e.g. item 3 - "Improving Crop Productivity and Resource.... "; It is common to write "Improving crop productivity and resource ..." in lower case.
Response 8: Thank you very much for your helpful comments. According to your request, we have already changed the writing style of all reference titles.
Round 2
Reviewer 1 Report
Comments and Suggestions for Authors
The manuscript has been revised substantially in light of the suggestions given earlier. Authors only need to include a few lines related to literature on phenology in the introduction section as well. Also, the ANOVA related results can be included as a Table for all parameters for more clarity, rather than presenting separately in each figure.
Author Response
Dear Reviewer
Thank you for your comments on our manuscript (agronomy-2710388) entitled "Optimized farmland mulching improves rainfed maize productivity by regulating soil temperature and phenology on the Loess Plateau in China". These comments are very helpful to the revision and improvement of our paper, and have important guiding significance for our research. We have carefully studied the comments and made corrections in the hope of approval. In the latest manuscript, revised sections are marked in red.
Reviewer #1:
The manuscript has been revised substantially in light of the suggestions given earlier. Authors only need to include a few lines related to literature on phenology in the introduction section as well. Also, the ANOVA related results can be included as a Table for all parameters for more clarity, rather than presenting separately in each figure.
Responses to Reviewer #1: Thank you very much for your recognition of this study. We have added text and literature on phenology in the Introduction section. Please Line 86—88. As follows:At the same time, as a key indicator influencing maize yield, the increase in soil temperature will shorten the phenological period of crops, ultimately leading to a reduction in maize production [23].Additionally, we have retabulated the three-factor analysis so that it can be clearly presented and expressed. Please Line 397, 428, 459, 492.
We believe that our study makes a significant contribution to the literature because we demonstrated that a flat plot of transparent film mulching with whole maize stalks could be an effective practice to extend growth and increase mulching spring maize yield and water use efficiency in the dryland of the Loess Plateau in China. Further, we believe that this paper will be of interest to the readership of your journal because consistent with your aims and scope, we provided a basis for further optimizing cultivation methods based on premature senescence, reducing the yield of film-mulched spring maize in rainfed agricultural areas in the Loess Plateau, and promoting the soil water efficiency and grain yield. Thank you again for your consideration of our revised manuscript.
Sincerely,
Haidong Lu
College of Agronomy, Northwest A&F University, Yangling 712100, China
Key Laboratory of Biology and Genetic Improvement of Maize in Arid Area of Northwest Region, Ministry of Agriculture, Yangling 712100, China
E-mail: [email protected]